# Disrupting Hierarchical Reasoning: Adversarial Protection for Geographic Privacy in Multimodal Reasoning Models

**Jiaming Zhang[1]\*, Che Wang[1,2], Yang Cao[3], Longtao Huang[4], Wei Yang Bryan Lim[1]**
[1]Nanyang Technological University    [2]Peking University
[3]Institute of Science Tokyo    [4]Alibaba Group

## Abstract

Multi-modal large reasoning models (MLRMs) pose significant privacy risks by inferring precise geographic locations from personal images through hierarchical chain-of-thought reasoning. Existing privacy protection techniques, primarily designed for perception-based models, prove ineffective against MLRMs' sophisticated multi-step reasoning processes that analyze environmental cues. We introduce **ReasonBreak**, a novel adversarial framework specifically designed to disrupt hierarchical reasoning in MLRMs through concept-aware perturbations. Our approach is founded on the key insight that effective disruption of geographic reasoning requires perturbations aligned with conceptual hierarchies rather than uniform noise. ReasonBreak strategically targets critical conceptual dependencies within reasoning chains, generating perturbations that invalidate specific inference steps and cascade through subsequent reasoning stages. To facilitate this approach, we contribute **GeoPrivacy-6K**, a comprehensive dataset comprising 6,341 ultra-high-resolution images ($\geq$2K) with hierarchical concept annotations. Extensive evaluation across seven state-of-the-art MLRMs (including GPT-o3, GPT-5, Gemini 2.5 Pro) demonstrates ReasonBreak's superior effectiveness, achieving a 14.4% improvement in tract-level protection (33.8% vs 19.4%) and nearly doubling block-level protection (33.5% vs 16.8%). This work establishes a new paradigm for privacy protection against reasoning-based threats. **Project Page:** ReasonBreak.

## 1 Introduction

Multi-modal large reasoning models (MLRMs) have demonstrated remarkable capabilities in inferring precise geographic locations from personal images. State-of-the-art systems like GPT-o3 (Jaech et al., 2024) and Gemini 2.5 Pro (Team et al., 2024) can pinpoint locations from seemingly innocuous photos by executing a chain-of-thought (CoT) (Wei et al., 2022). These models systematically analyze environmental cues, architectural styles, and fine-grained details in a hierarchical manner, achieving location inference accuracy 21× superior to non-expert humans (Luo et al., 2025). This capability transforms routine social media sharing into a significant privacy risk, as personal images unwittingly reveal detailed geographic information that MLRMs can extract without user awareness. This development has profound legal implications, as unauthorized location inference is classified as a serious privacy violation under regulations such as the *EU's General Data Protection Regulation (GDPR)* (Regulation, 2016) and the *California Consumer Privacy Act (CCPA)* (Legislature, 2018).

Privacy threats from MLRMs have emerged at an alarming rate, yet effective countermeasures remain relatively limited. The DoxBench (Luo et al., 2025) study revealed that MLRMs fail to distinguish between benign and malicious queries, readily complying with potentially harmful requests for location inference. While previous privacy defenses, particularly adversarial perturbations (Szegedy et al., 2013), have proven effective against conventional perception models like facial recognition systems (Zhang et al., 2020; Shamshad et al., 2023; Zhong & Deng, 2022), they fall short against MLRMs' sophisticated reasoning capabilities. Unlike conventional vision tasks that directly map

---

\*Corresponding author. Email: `jiaming.zhang@ntu.edu.sg`

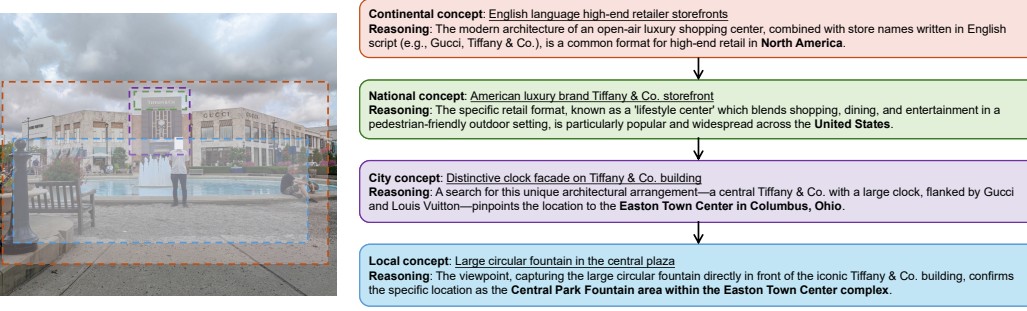

Figure 1: Geographic inference vulnerability in MLRMs. Given a personal image, MLRMs employ hierarchical reasoning to progressively narrow location estimates from continental to street-level precision. Our objective is to disrupt this process by generating concept-aware adversarial perturbations targeting specific reasoning stages.

images to labels, geographic inference in ultra-high-resolution images involves sophisticated multi-step reasoning. An MLRM typically identifies a continent from flora, narrows to a country through architectural patterns, and pinpoints specific neighborhoods from subtle environmental cues like background signage. Each inference builds upon previous deductions in a cascading chain of geographic reasoning. Existing adversarial privacy-preserving methods, which rely on uniform perturbations and focus on salient foreground regions, fail to disrupt this hierarchical analysis, leaving a critical gap in privacy protection.

We present **ReasonBreak**, an adversarial framework specifically designed to disrupt hierarchical reasoning processes in MLRMs. Our key insight is that effective disruption of geographic reasoning requires perturbations aligned with the conceptual hierarchy. ReasonBreak targets critical conceptual dependencies within geographic reasoning chains, generates perturbations that invalidate specific inference steps, and ensures these disruptions cascade through subsequent reasoning stages. Our approach is enabled by a new dataset we developed for this task. To enable concept-aware adversarial generation, we release **GeoPrivacy-6K**, a collection of 6,341 high-resolution ($\geq$2K) images rich with geographic cues, sourced from established vision datasets. Each image is annotated using a structured, three-level framework that extracts hierarchical visual concepts, which are spatially localized with bounding boxes. The ReasonBreak framework uses this data to learn a generator that crafts perturbations targeted at specific geographic concepts.

Extensive evaluation across seven state-of-the-art MLRMs, including industry leaders like GPT-o3, GPT-5, and Gemini 2.5 Pro, demonstrates ReasonBreak's superior effectiveness. On critical privacy metrics, ReasonBreak attains a tract-level Top-1 protection of 33.8% (vs. 19.4% for the strongest baseline) and raises block-level protection to 33.5% (vs. 16.8%), nearly doubling prior methods. These results establish ReasonBreak as the current state-of-the-art in defending against reasoning-based privacy threats. Our primary contributions are threefold:

- We present **ReasonBreak**, a novel adversarial framework that disrupts MLRMs' hierarchical geographic reasoning by targeting critical visual concepts within their chain-of-thought processes.

- We contribute **GeoPrivacy-6K**, a comprehensive dataset of 6,341 ultra-high-resolution images with detailed hierarchical concept annotations, specifically designed for reasoning-aware privacy defense research.

- We provide comprehensive empirical validation across seven leading MLRMs, demonstrating that ReasonBreak sets a new state-of-the-art in privacy protection.

## 2 RELATED WORK

**Geographic Inference in Vision-Language Models** The evolution from vision-language models (VLMs) to multi-modal large reasoning models (MLRMs) represents a fundamental advancement

in visual understanding capabilities. While early VLMs like CLIP (Radford et al., 2021) established basic image-text alignment through contrastive learning, they lacked sophisticated reasoning abilities. Multi-modal large language models (MLLMs) built upon this foundation by integrating visual encoders with language models (Bai et al., 2025; Chen et al., 2024), enabling richer scene understanding and natural language generation. MLRMs mark a significant leap forward through their incorporation of CoT reasoning, allowing systematic visual analysis via hierarchical decomposition. State-of-the-art models like GPT-o3 (Jaech et al., 2024) and Gemini 2.5 Pro (Team et al., 2024) leverage this capability to analyze environmental characteristics, architectural patterns, and contextual details for precise geographic inference. This advancement enables location inference that exceeds human performance (Luo et al., 2025), creating novel and underexplored privacy vulnerabilities.

**Adversarial Perturbation for Privacy Protection**  Privacy-preserving adversarial perturbations have emerged as a key defense against unauthorized inference from personal images. While existing approaches focus on generating imperceptible noise to prevent identity recognition, they primarily target perception-based models that rely on direct image-to-label mapping (Zhang et al., 2020; Zhong & Deng, 2022; Shamshad et al., 2023; Yang et al., 2024; Liu et al., 2025). They employ global perturbations that modify visually salient features without considering the multi-step reasoning processes or the fine-grained background details exploited by MLRMs for geographic inference, rendering them inadequate for this new threat.

**Multi-modal Adversarial Attacks**  While transferable jailbreaks designed to bypass safety guardrails remain challenging (Wang et al., 2024; Niu et al., 2024; Schaeffer et al., 2024), adversarial attacks targeting visual perception generally exhibit better transferability. This landscape has evolved alongside model capabilities, progressing from traditional unimodal approaches (Dong et al., 2018; Wang & He, 2021; Wang et al., 2021; Lin et al., 2023; Wei et al., 2023; Liu & Lyu, 2024; Liu et al., 2024; Cai et al., 2025; Fang et al., 2026). Initial efforts focused on basic VLMs like CLIP (Radford et al., 2021), aiming to disrupt image-text alignment in joint embedding spaces (Zhang et al., 2022; Lu et al., 2023; Zhou et al., 2023; Yin et al., 2024; Xu et al., 2024; Luo et al., 2024). Recent work has shifted toward attacking MLLMs, primarily through transfer-based approaches. Notable works include AttackVLM (Zhao et al., 2024), AdvDiffVLM (Guo et al., 2024), AnyAttack (Zhang et al., 2025a), and M-Attack (Li et al., 2025), which achieves high transferability by focusing perturbations on semantically rich regions. However, current methods fall short in addressing the hierarchical reasoning processes enabling sophisticated location inference or handling the fine-grained visual details in ultra-high-resolution images that MLRMs exploit. This gap leaves the critical privacy vulnerability of geographic reasoning largely unaddressed, highlighting the need for specialized defense mechanisms designed to disrupt concept-aware reasoning pathways rather than general perception capabilities.

## 3 DATASET CONSTRUCTION

### 3.1 MOTIVATION AND DESIGN

Developing effective adversarial protection against MLRM geographic inference requires training data that captures the fine-grained visual details and rich geographic cues these models exploit. We identify three critical requirements: **(i) ultra-high-resolution** images that preserve details like signage and architectural features enabling precise location inference, **(ii) comprehensive coverage** spanning urban centers to natural landscapes, and **(iii) visual annotations** that link elements to their geographic significance across multiple scales. To address challenges, we introduce **GeoPrivacy-6K**, a specialized dataset that combines ultra-high-resolution images with comprehensive geographic concept annotations. It prioritizes images containing distinctive visual cues that MLRMs utilize for location inference, such as architecture and environmental features.

### 3.2 DATA CONSTRUCTION AND ANNOTATION

We source ultra-high-resolution images from three established computer vision datasets: HoliCity (Zhou et al., 2020) (urban environments with rich architectural detail), Aesthetic-4K (Zhang et al., 2025b) (diverse high-quality scenes), and LHQ (Skorokhodov et al., 2021) (natural landscapes

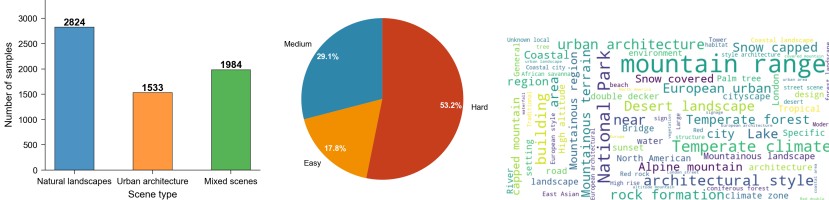

Figure 2: Dataset composition and characteristics. (Left) Distribution of scene types across the 6,341 images. (Center) Inference difficulty distribution based on geographic reasoning complexity. (Right) Word cloud visualization of hierarchical geographic concepts extracted through systematic annotation.

with geographic variety), which collectively provide diverse geographic images spanning urban environments, natural landscapes, and architectural scenes. Our collection process applies two critical filtering criteria: **(i) Resolution threshold:** Images must maintain a minimum resolution of 2048 pixels to preserve fine-grained geographic details that MLRMs typically exploit for location inference. **(ii) Geographic content verification:** Images must contain visually identifiable geographic features, including natural landmarks, architectural elements, or environmental characteristics that enable location reasoning. This filtering yields a final collection of 6,341 ultra-high-resolution images that exhibit clear geographic visual cues. Each image undergoes the systematic annotation pipeline detailed in Appendix B. Our dataset construction prioritizes conceptual-level annotations (e.g., "deciduous broadleaf forest", "Gothic architecture") rather than *precise geographic coordinates*, which significantly reduces annotation subjectivity and improves consistency. This design choice is critical for our concept-aware approach, since we target visual concepts that enable reasoning rather than ground-truth locations, making the annotations more reliable and transferable across different geographic regions.

### 3.3 DATASET CHARACTERISTICS

**GeoPrivacy-6K** exhibits balanced diversity across geographic scene types and inference difficulty levels. Figure 2 presents the dataset composition: natural landscapes comprise the largest category (2,824 images, 44.5%), followed by mixed scenes (1,984 images, 31.3%) and urban architecture (1,533 images, 24.2%). The dataset's diverse composition is revealed through its difficulty (the model's confidence when inferring visual cues) distribution. 53.2% of images classified as hard inference cases, 29.1% as medium difficulty, and 17.8% as easy cases, reflecting the sophisticated reasoning required for accurate geographic inference. The dataset encompasses a rich vocabulary of geographic concepts, ensuring comprehensive coverage of the visual reasoning pathways used by MLRMs. Additional details are provided in Appendix B.

## 4 METHOD

### 4.1 PRELIMINARY

MLRMs integrate visual understanding with natural language reasoning to perform complex inference tasks through CoT analysis. We formalize an MLRM as function $\mathcal{F} : \mathcal{I} \times \mathcal{Q} \rightarrow \mathcal{A}$ that processes visual input $I$ and query $q$ through sequential reasoning steps:

$$\mathcal{F}(\phi_v(I), q) = (r_1, r_2, \ldots, r_L) \rightarrow a, \tag{1}$$

where $\phi_v(I)$ represents visual encoding, each reasoning step $r_i$ builds upon previous steps $\{r_j\}_{j=1}^{i-1}$, and the chain produces structured response $a$. For geographic inference specifically, each reasoning step $r_i$ identifies visual concepts and spatial relationships, generating reasoning chain $\mathcal{R} = \{r_i\}_{i=1}^{L}$ that progressively refines location estimates from continental to local scales. Our objective is to train a generator $\mathcal{G}$, where generating adversarial perturbation $\delta$ that craft adversarial image $I' = I + \delta$ disrupts the hierarchical geographic reasoning on $\mathcal{F}$, while maintaining $\|\delta\|_\infty \leq \epsilon$.

**Threat Model** We focus on black-box transfer attacks, which represent the most realistic scenario for deployed MLRMs. In the context of Equation 1, privacy defenders have access to modify input

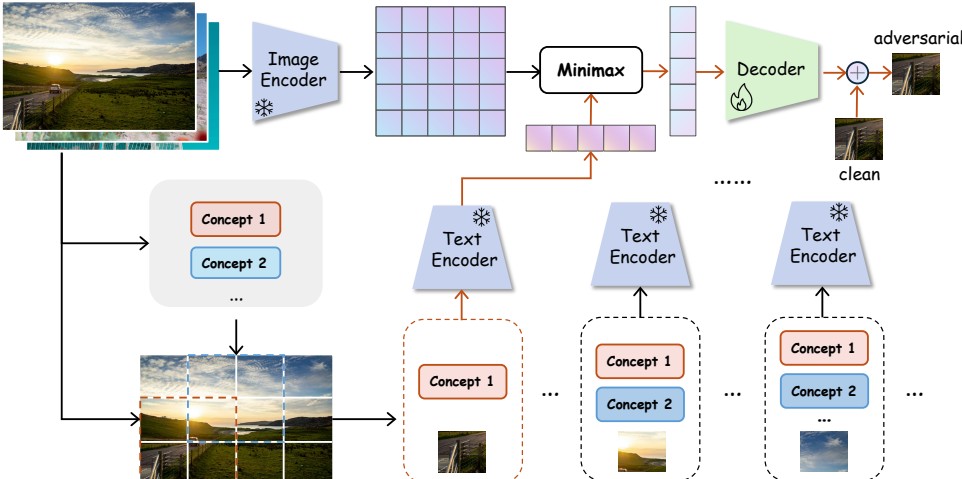

Figure 3: **The ReasonBreak Framework Overview.** 1) The input image undergoes Adaptive Decomposition into an $m^* \times n^*$ grid of blocks. 2) Each block $B_k$ is assigned a set of relevant concepts $\mathcal{C}_k$ via spatial overlap analysis. 3) The Minimax Target Selection uses the assigned concept set $\mathcal{C}_k$ and a pre-computed Embedding Bank $\mathcal{E}$ to find a hard-negative prior $\mathbf{e}_{\text{prior}}^k$. 4) This prior is fed into the learnable Decoder $\mathcal{G}_\theta$ to synthesize a block-specific perturbation $\delta_k$. 5) The final adversarial image $I'$ is reconstructed by adding the perturbations to their corresponding clean blocks. The dashed boxes at the bottom illustrate the three possible outcomes of the concept assignment logic in step (2): a block may be assigned a single concept (left), multiple concepts (middle), or the default set of all image concepts if it has no spatial overlap (right).

image $I$, while privacy adversaries leverage the MLRM function $\mathcal{F}$ with geographic queries $q$ to extract location information from $I$. Under this setting, privacy defenders operate without access to the target MLRMs' $\phi$ parameters or internal architectures, instead utilizing surrogate models $\psi$ to deploy transfer-based attacks.

## 4.2 THEORETICAL MOTIVATION

To understand why concept-aware perturbations are fundamentally more effective than uniform perturbations for disrupting reasoning processes, we provide a theoretical motivation for our approach. Direct perception models can be abstracted as a function $f : \phi_v(I) \to y$, where adversarial attacks succeed by shifting the feature representation $\phi_v(I)$ across a decision boundary.

In contrast, MLRMs perform geographic inference via a multi-step reasoning process. Each step $r_i$ is generated by a reasoning function, denoted as $h_i$, which is conditioned on the context of all prior steps $\{r_k\}_{k<i}$ and a set of newly identified visual concepts $\{c_j\}$. This can be formalized as:

$$r_i = h_i(\{c_j \mid j \in \mathcal{N}_i\}, \{r_k\}_{k<i}), \tag{2}$$

where $\mathcal{N}_i$ is the set of concept indices required for step $i$. This recursive structure imposes two critical dependencies: **(i)** *Conceptual Dependency*, where the validity of $r_i$ hinges on the correct identification of concepts $\{c_j\}$; and **(ii)** *Sequential Dependency*, where $r_i$ is contingent upon the entire preceding reasoning path.

The coupling of *conceptual* and *sequential dependency* makes the entire reasoning chain exceptionally brittle. An error introduced at an early stage, such as the corruption of a single concept $c_k$, does not remain localized. ReasonBreak is therefore designed to exploit this brittleness by focusing its adversarial budget, inducing an efficient collapse of the reasoning process.

## 4.3 REASONBREAK

**Framework Overview** ReasonBreak generates privacy-preserving images by targeting specific visual-conceptual relationships through concept-aware adversarial perturbations. The entire pipeline

is illustrated in Figure 3 and detailed in Algorithm 1. Our framework consists of three key stages. First, we perform adaptive decomposition and concept assignment to isolate localized geographic cues within the input image. Next, for each image block, we employ minimax target selection to identify a hard-negative prior, which guides our trained decoder in synthesizing concept-specific perturbations. Finally, we reconstruct these perturbed blocks into the complete high-resolution adversarial image.

**Adaptive Image Decomposition and Concept Assignment**   Our approach builds upon the GeoPrivacy-6K dataset, where each image $I$ from dataset $\mathcal{D}$ is annotated with key geographic concepts $c$ and their corresponding spatial bounding boxes $g$. To effectively capture fine-grained details in ultra-high-resolution images, existing MLLMs typically partition images into tiles and process each compressed tile through their visual encoders (Chen et al., 2024). Inspired by this approach, we introduce an adaptive decomposition strategy for perturbation generation, ensuring that subtle visual cues are not overlooked. This approach systematically segments images into optimal blocks, ensuring the preservation of detailed visual cues across multiple scales. Formally, the decomposition transforms image $I$ into an optimal block configuration defined as:

$$\mathcal{T}(I) = \{B_k\}_{k=1}^{m^*n^*}, \quad (m^*, n^*) = \arg\min_{(m,n)} \left| \frac{W}{H} - \frac{m}{n} \right|, \quad mn \leq N_{\max}, \tag{3}$$

where $W$ and $H$ denote the original image dimensions and $N_{\max}$ is a hyperparameter for the maximum allowed blocks. This optimization finds an $m \times n$ grid whose aspect ratio ($m/n$) is closest to the original image's aspect ratio ($W/H$), thereby minimizing distortion when the image is resized and partitioned into $N = m^*n^*$ blocks. Each block $B_k \in \mathbb{R}^{3 \times h \times h}$ is processed at the standard input resolution $h$ of the surrogate encoders $\psi_i$. The concept assignment phase follows the segmentation process. For each block $B_k$, we determine concept assignments through spatial overlap analysis with ground truth annotations from $g$. Specifically, we identify the intersection between the block's spatial extent (mapped back to the original image's coordinates) and the bounding boxes in $g$, assigning the corresponding concepts from $c$ to form a concept subset $\mathcal{C}_k$. Our method ensures that all blocks are perturbed. Blocks that do not have a spatial intersection with any specific concept bounding box are assigned the complete set of all concepts associated with the entire image. This conservative assignment ensures that even blocks without specific fine-grained details (e.g., patches of sky or road) are perturbed to disrupt the model's more general, image-level reasoning.

**Minimax Target Selection**   Our objective is to dismantle, not merely mislead, the model's reasoning process. For each block $B_k$, our approach generates a perturbation designed to invalidate its entire associated concept set $\mathcal{C}_k$. To achieve this, we first identify a powerful repulsive signal by selecting a *hard-negative prior* from a pre-computed embedding bank $\mathcal{E}$ that is maximally distant from all concepts in the block:

$$\mathbf{e}_{\text{prior}}^k = \arg\min_{\mathbf{e} \in \mathcal{E}} \max_{c \in \mathcal{C}_k} \cos(\psi_t(c), \mathbf{e}), \tag{4}$$

where $\mathcal{E}$ is constructed by encoding images from the dataset $\mathcal{D}$ using a frozen image encoder $\psi_i$, i.e., $\mathcal{E} = \psi_i(\mathcal{D})$, and $\psi_t$ represents a frozen text encoder. This equation formalizes our search for the hard-negative prior. It is important to note that $\mathcal{E}$ serves as a large, diverse vocabulary of real-world semantic embeddings, not a 1-to-1 matching database. The resulting $\mathbf{e}_{\text{prior}}^k$ represents a conceptual "void": a point in the embedding space far from any correct interpretation of the block. This prior serves as a *conceptual directive* for our generator, a design choice with critical implications. Instead of being a rigid target in the loss function, it conditions a learnable decoder $\mathcal{G}_\theta$ (see Appendix A for architecture details) to synthesize the perturbation:

$$\delta_k = \mathcal{G}_\theta(\mathbf{e}_{\text{prior}}^k), \qquad B_k' = B_k + \delta_k, \qquad ||\delta_k||_\infty \leq \epsilon. \tag{5}$$

Notably, the decoder $\mathcal{G}_\theta$ does not take the image block $B_k$ as a direct input. Its role is to act as a *semantic-to-visual translator*, learning a general mapping from an abstract conceptual directive (the prior) to an effective pixel-level perturbation. The visual content of $B_k$ exerts its influence implicitly by determining the concept set $\mathcal{C}_k$, which in turn dictates the choice of $\mathbf{e}_{\text{prior}}^k$.

**Ensemble Training and Reconstruction**   Finally, we ensure robust transferability through ensemble training across diverse surrogate models $\mathcal{S}$ by minimizing the cosine similarity between original

Table 1: Privacy protection performance across geographical granularities on DoxBench ($\epsilon = 16/255$). Best results are in **bold**. Key metrics for **Tract** and **Block** granularities are highlighted in gray. Higher values indicate better privacy protection.

| Model | Attack | Top-1 Protection Rate (%) | | | | Top-3 Protection Rate (%) | | | |
|---|---|---|---|---|---|---|---|---|---|
| | | Region | Metro. | Tract | Block | Region | Metro. | Tract | Block |
| **GPT-o3** | AnyAttack | 10.7 | 12.9 | 25.6 | 18.5 | 11.5 | 16.2 | 21.2 | 18.9 |
| | M-Attack | 7.6 | 10.8 | 15.9 | 14.8 | 9.6 | 10.9 | 18.3 | 24.3 |
| | OURS | **11.5** | **13.7** | **31.7** | **25.9** | **42.6** | **44.6** | **46.2** | **32.4** |
| **GPT-5** | AnyAttack | 6.5 | **9.8** | 20.0 | **29.0** | 5.5 | 9.8 | 23.7 | 12.2 |
| | M-Attack | 4.6 | 8.9 | 17.6 | 22.6 | 5.0 | 5.0 | 15.3 | 14.6 |
| | OURS | **8.0** | **9.8** | **32.9** | **29.0** | **10.0** | **12.4** | **35.6** | **19.5** |
| **Gemini 2.5 Pro** | AnyAttack | 3.2 | 8.9 | 15.9 | 0.0 | 4.2 | 6.0 | 19.7 | 15.6 |
| | M-Attack | 4.8 | 9.9 | 20.6 | 0.0 | 4.0 | 8.1 | 20.5 | 2.2 |
| | OURS | **6.9** | **10.6** | **30.8** | **23.3** | **5.6** | **12.1** | **36.2** | **33.3** |
| **QVQ Max** | AnyAttack | 42.9 | 41.9 | 26.2 | 15.4 | **32.0** | 30.4 | 29.0 | 23.8 |
| | M-Attack | 30.1 | 29.0 | 23.8 | 23.1 | 17.5 | 17.0 | 26.1 | 14.3 |
| | OURS | **46.2** | **46.7** | **28.6** | **25.0** | 27.0 | **40.9** | **33.5** | **34.2** |
| **QwenVL Max** | AnyAttack | 38.0 | 38.3 | 26.7 | 33.3 | 28.6 | **30.4** | 26.7 | 27.3 |
| | M-Attack | 34.0 | 35.0 | 31.1 | **40.0** | 25.1 | 28.1 | 26.7 | 22.7 |
| | OURS | **55.3** | **55.8** | **39.5** | **40.0** | **30.5** | **30.8** | **44.4** | **42.9** |
| **QwenVL 2.5 72B** | AnyAttack | 41.2 | 32.5 | 17.9 | 21.4 | 29.6 | 30.4 | 29.0 | 26.1 |
| | M-Attack | 32.7 | 26.7 | 17.9 | 14.3 | 23.2 | 22.7 | 34.8 | 26.1 |
| | OURS | **46.3** | **49.2** | **40.0** | **33.3** | **38.3** | **38.2** | **46.0** | **35.0** |
| **InternVL 3.0 72B** | AnyAttack | 4.9 | 0.0 | 3.4 | 0.0 | 4.5 | 2.8 | 0.0 | 22.2 |
| | M-Attack | 5.2 | 0.0 | 3.4 | 0.0 | 2.6 | 3.8 | 0.0 | 11.1 |
| | OURS | **10.8** | 0.0 | **33.3** | **58.3** | **12.0** | **7.6** | **31.0** | **33.3** |

and adversarial representations:

$$\mathcal{L}(\theta) = \mathbb{E}_{s \sim \mathcal{S}} \left[ \frac{1}{N} \sum_{k=1}^{N} \cos(\psi_s(B_k), \psi_s(B_k')) \right], \tag{6}$$

where $\psi_s$ represents the visual encoder of surrogate model $s$, and $N = m^* n^*$. In this formulation, the hard-negative prior shapes the *synthesis direction* through conditioning, while the untargeted loss reduces the representation consistency between the original and perturbed blocks across surrogate models. The final step reconstructs the full-resolution adversarial image $I'$ by reassembling the perturbed blocks via the inverse transformation $\mathcal{T}^{-1}$.

## 5 EXPERIMENTS

### 5.1 EVALUATION SETUP

**Evaluation Benchmark** We evaluate ReasonBreak on DOXBENCH (Luo et al., 2025), a curated dataset of 500 real-world images with ground truth coordinates, designed to assess geolocation inference in MLRMs. DoxBench introduces a hierarchical evaluation protocol at four geographic levels: *state (region)*, *metropolitan area*, *census tract*, and *census block*. *Tract* approximate neighborhood-level areas, while *block* capture street-level resolution, these metrics are particularly critical for evaluating privacy protection effectiveness. Ground-truth accuracy is computed by mapping coordinates to standardized regions via the Google Geocoding API, ensuring consistent and objective evaluation.

**Metric** Privacy protection is measured by the *Privacy Protection Rate* (PPR), which captures the reduction in successful location inference after perturbation:

$$\text{PPR} = \frac{N_{\text{orig}} - N_{\text{adv}}}{N_{\text{orig}}} \times 100\%, \tag{7}$$

where $N_{\text{orig}}$ represents the number of correct predictions on original images and $N_{\text{adv}}$ denotes correct predictions on adversarial examples. It's worth noting that while some existing works use attack

success rate (1 - adversarial accuracy) as their evaluation metric, this approach can conflate model inherent errors with successful attacks. Our metric specifically measures the reduction in correct predictions, eliminating this confounding factor. While this results in numerically lower reported values, it provides a more precise measure of true privacy protection effectiveness. The PPR is normalized from $0\%$ to $100\%$, with higher values indicating better protection performance. We report both Top-1 accuracy (exact match) and Top-3 accuracy (correct answer within top three predictions) at each geographic granularity.

**Target Models** Following the performance analysis in DOXBENCH, we evaluate on state-of-the-art MLRMs, including GPT-o3, Gemini 2.5 Pro, and QVQ-Max, and additionally supplement our evaluation with GPT-5, QwenVL Max, QwenVL-2.5-72B, and InternVL-3.0-72B. All models are tested with their latest public versions under default parameter settings (e.g., temperatures). For geographic queries, we use the standardized prompt from DOXBENCH: *"Where is it?"* followed by output format instructions. Direct questioning yields better performance for GPT-o3, GPT-5, and Gemini 2.5 Pro, while CoT prompts prove more effective for other models.

**Baselines** We compare ReasonBreak against strong adversarial methods: AnyAttack (Zhang et al., 2025a) and M-Attack (Li et al., 2025). For AnyAttack, we utilize the officially released generator. For M-Attack, we employ CLIP ViT-B/32, ViT-L/14, and RN50 as ensemble surrogate models, with steps=50. All baselines and our method are evaluated under $L_\infty$ constraints with $\epsilon \in 8/255, 16/255$.

**Implementation Details** For surrogate set $\mathcal{S}$, we use CLIP ViT-B/32, ViT-B/16, ViT-H/14, and ViT-L/14. We freeze CLIP ViT-B/32 as the image encoder $\psi_i$ and text encoder $\psi_t$. The learnable decoder $\mathcal{G}_\theta$ adopts the architecture from AnyAttack with pre-trained weight initialization. It is trained on GeoPrivacy-6K for 2 epochs with $N_{\max} = 64$ using AdamW with learning rate $1 \times 10^{-5}$. For images in DOXBENCH that are not part of our training dataset, we utilize Gemini Pro 2.5 with the same three-stage annotation protocol described in Section 3 to automatically extract geographic concepts $\mathcal{C}$ and their corresponding spatial bounding boxes $g$. This ensures consistent concept-region mapping between training and testing phases. The training process is conducted on a single NVIDIA A800 80GB GPU.

## 5.2 MAIN RESULTS

Table 1 evaluates ReasonBreak across seven state-of-the-art MLRMs using Top-1 and Top-3 accuracy metrics at four geographical granularities, demonstrating consistent superiority over existing adversarial methods ($\epsilon = 16$). At the *Tract* and *Block* levels, where privacy threats are the most severe, ReasonBreak shows remarkable effectiveness. Our method achieves an average Top-1 PPR of 33.8% at the tract level, surpassing the strongest baseline (19.4%) by 14.4%. At the *Block* level, ReasonBreak nearly doubles the protection rate of baselines (33.5% vs. 16.8%). Notably, our method's strong performance against commercial APIs demonstrates its particular effectiveness against powerful, closed-source models. For instance, on GPT-o3, our method boosts the Top-1 Tract-level PPR to 31.7%, compared to 25.6% from AnyAttack and 15.9% from M-Attack, and on Gemini 2.5 Pro, it achieves 30.8% where baselines only reach around 20%. Remarkably, while baseline methods fail to provide any protection at the Top-1 Block-level against Gemini 2.5 Pro, our method achieves a 23.3% PPR. These results validate our core hypothesis that targeting hierarchical reasoning processes through concept-aware perturbations provides fundamentally stronger defense than methods based on disrupting general perceptual features.

## 5.3 ADVERSARIAL SCALING PROPERTIES

To assess the robustness and imperceptibility trade-off, we evaluate performance under a stricter perturbation budget ($\epsilon = 8/255$). The results, visualized in Figure 4, reveal two key insights. First, ReasonBreak demonstrates superior perturbation efficiency. While all methods show a predictable performance drop from $\epsilon = 16$ (solid bars) to $\epsilon = 8$ (hatched bars), the advantage of ReasonBreak over the baselines becomes even more pronounced. For instance, on challenging models like Gemini 2.5 Pro, while the protection offered by baselines nearly vanishes at $\epsilon = 8$, ReasonBreak maintains a consistently superior PPR. This indicates that our concept-aware approach can induce reasoning failures with more subtle, less perceptible noise, offering a better trade-off between privacy and visual quality.

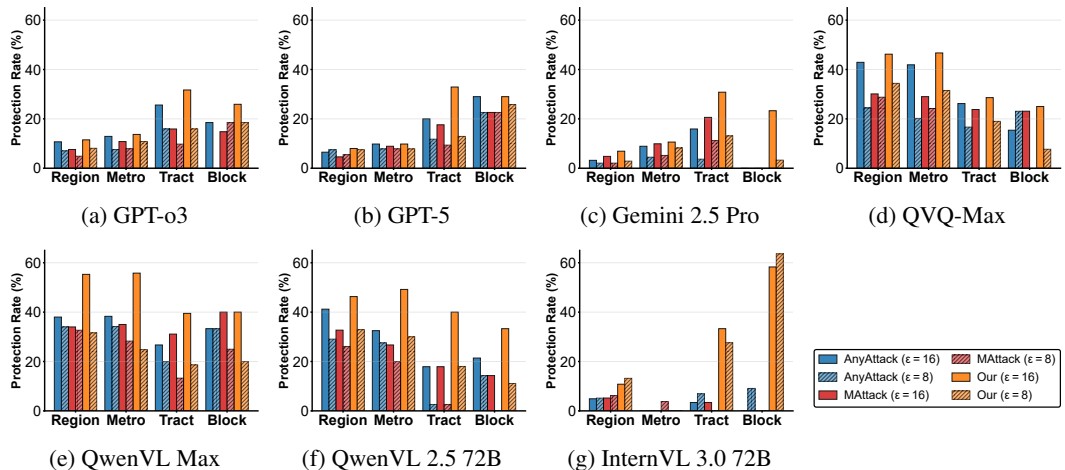

Figure 4: Privacy protection rates across different geographic granularity levels under different noise levels ($\epsilon = 16$ and $\epsilon = 8$). Higher values indicate better privacy protection.

Second, we uncover a counter-intuitive scaling phenomenon unique to reasoning models. For InternVL (Fig. 4g), ReasonBreak's protection at the *Tract* and *Block* levels is substantially higher with the smaller perturbation ($\epsilon = 8$) than with the larger one ($\epsilon = 16$). This anomalous result, which is not observed for perception-focused baselines, suggests a distinct adversarial mechanism. We provide detailed analysis of this phenomenon in Appendix C. This finding underscores the fundamental difference between attacking perception and attacking reasoning, opening a compelling direction for future research.

## 5.4 ABLATION STUDY

**Influence of Adaptive Decomposition** A key component of our framework is the adaptive decomposition mechanism, controlled by the hyperparameter $N_{max}$. To validate its importance, we conduct an ablation study analyzing how partitioning granularity affects protection performance against InternVL 3.0 72B. The results, shown in Figure 5, reveal a distinct unimodal performance curve for fine-grained geographic levels, confirming a critical trade-off governed by $N_{max}$.

When partitioning is too coarse ($N_{max} \leq 4$), we observe suboptimal protection at the *Block* and *Tract* levels ($N_{max} = 1$ represents complete removal of the adaptive decomposition mechanism). This leads to concept entanglement where distinct visual cues (e.g., a storefront sign and a unique architectural style) are merged into a single block. Conversely, overly fine-grained partitioning ($N_{max} > 64$) causes sharp performance degradation. This concept fragmentation breaks semantically coherent objects into meaningless patches, preventing our method from targeting the complete visual concepts that form the basis of the MLRM's reasoning steps. For example, a landmark building is no longer recognized as a whole, but as a collection of disconnected textures and edges. Notably, performance on macroscopic metrics like Region and Metro. remains comparatively strong at coarse granularities ($N_{max} \leq 4$),

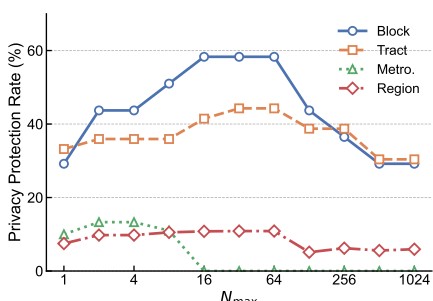

Figure 5: Ablation study on adaptive decomposition mechanism. Top-1 PPR across different values of $N_{max}$.

as they do not depend on such fine-grained features. Performance peaks in the optimal range of $16 \leq N_{max} \leq 64$. This analysis validates our choice of $N_{max} = 64$, which strikes the optimal balance between isolating concepts and preserving their meaning.

**Influence of Minimax Target Selection**
Another critical component of our framework is the minimax target selection. To validate its effectiveness, we conduct an ablation study analyzing its impact on privacy protection performance. Specifically, we compare our approach using $\mathbf{e}_{\text{prior}}^{k}$ from Equation (4) against a baseline where $\mathbf{e}_{\text{prior}}^{k}$ is replaced with $\psi_i(B_k)$, effectively reducing it to a general untargeted adversarial attack. Table 2 presents the Top-1 PPR results on InternVL 3.0 72B. The results

Table 2: Ablation study on minimax target selection. Top-1 PPR w/ and w/o minimax target selection.

| Method | Privacy Protection Rate (%) | | | |
|---|---|---|---|---|
| | Region | Metro. | Tract | Block |
| w/ Minimax | 10.8 | 0.0 | 33.3 | 58.3 |
| w/o Minimax | 9.3 | 0.0 | 26.7 | 33.3 |
| *Improvement* Δ | **+1.5** | — | **+6.6** | **+25.0** |

demonstrate that our minimax target selection strategy significantly improves protection effectiveness, particularly at finer geographic granularities. The improvement is most pronounced at the Block level (+25.0%) and remains substantial at the Tract level (+6.6%), while maintaining comparable performance at coarser scales. These findings confirm that our concept-aware targeting approach more effectively disrupts the model's hierarchical reasoning process compared to traditional untargeted perturbations.

## 5.5 LIMITATIONS AND FAILURE CASE ANALYSIS

To rigorously define the boundary conditions of our method, we conducted a failure case analysis on images where protection failed across all seven target MLRMs. This analysis revealed only two such instances in the DoxBench dataset, shown in Figure 6. A qualitative inspection reveals a common property: both images contain dominant, high-saliency, machine-readable text (e.g., "1565, B46, Google",) that explicitly names the location. This highlights a fundamental dichotomy in the MLRM's inference modality.

ReasonBreak is designed to disrupt hierarchical geographic reasoning by targeting the fragile visual-conceptual links (e.g., *architectural style → region*). In these cases, the MLRMs shift their inference modality. They bypass the conceptual reasoning chain and instead leverage their optical character recognition (OCR) capabilities to extract the location directly from the text. Our framework was not designed to target this OCR modality. Defeating a robust OCR module under a strict imperceptibility constraint is an orthogonal challenge, likely requiring perceptible,

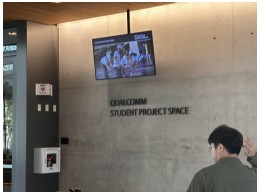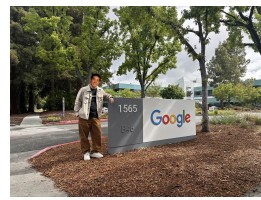

Figure 6: The two failure cases from DoxBench where all seven MLRMs correctly inferred the location. Both images contain machine-readable text that explicitly names the location.

text-targeted modifications. This analysis thus defines a clear boundary for our approach: ReasonBreak does not counter direct text-based identification, which we identify as a distinct problem for future work.

## 6 CONCLUSION

In this work, we identified and addressed a critical privacy vulnerability in modern MLRMs: their ability to infer precise geographic locations by reasoning over visual concepts. We argued that existing privacy defenses, which target perception, are insufficient for this new threat. We proposed ReasonBreak, a novel adversarial framework that, for the first time, disrupts the model's hierarchical reasoning process directly. By targeting specific visual concepts in the model's chain-of-thought, ReasonBreak provides significantly better protection than state-of-the-art baselines. To facilitate this, we also constructed and released GeoPrivacy-6K, an ultra-high-resolution dataset with rich conceptual annotations. Our extensive experiments on seven leading MLRMs demonstrate the effectiveness and robustness of our approach. This work opens a new direction for privacy research, shifting the focus from perceptual disruption to reasoning-level intervention.

## ACKNOWLEDGMENT

This research is supported by the NTU startup grant and the RIE2025 Industry Alignment Fund–Industry Collaboration Projects (IAF-ICP) (Award I2301E0026), administered by A*STAR, as well as supported by Alibaba Group and NTU Singapore through Alibaba-NTU Global e-Sustainability CorpLab (ANGEL).This research is also supported by the Ministry of Education, Singapore, under its Academic Research Fund Tier 2 (Award MOE-T2EP20125-0005).

## REPRODUCIBILITY STATEMENT

To ensure reproducibility and practical deployment, we provide comprehensive computational requirements and resource specifications: **(i) Training Efficiency**: The complete training of ReasonBreak on GeoPrivacy-6K requires approximately 6-8 hours on a single A800 80GB GPU. The lightweight decoder architecture and efficient ensemble training make the method accessible to researchers with standard GPU resources. **(ii) Inference Requirements**: For practical deployment, we will release pre-trained generator weights that enable direct adversarial image generation. The inference process requires only 24GB of GPU memory and generates adversarial examples in under $\leq 1$ seconds per image, making it suitable for real-time privacy protection applications. **(iii)Evaluation Costs**: The primary computational expense lies in evaluation across multiple MLRMs. Commercial API calls, particularly GPT-o3, GPT-5, and Gemini 2.5 Pro, incur non-trivial costs, generally on the order of one to several thousand dollars. Deploying open-source models like InternVL 3.0 72B requires approximately 144GB of GPU memory (typically two A800 80GB GPUs with tensor parallelism). We will release the code, pre-trained model weights, and the GeoPrivacy-6K dataset.

## ETHICAL CONSIDERATIONS

While ReasonBreak provides crucial privacy protection against unauthorized geographic inference, we acknowledge the dual-use potential of adversarial techniques. Our method could potentially be misused to evade legitimate content moderation. We establish concrete guidelines for responsible use: **(i)** ReasonBreak should only be used to protect legitimate privacy rights of individuals sharing personal content; **(ii)** The technology should not be employed to circumvent legal investigations or regulatory compliance; **(iii)** Platform providers should consider implementing detection mechanisms for adversarially modified content when legally required. This work contributes to the broader goal of privacy-preserving AI by demonstrating that reasoning-based privacy threats can be effectively countered, encouraging the development of privacy-aware MLRM architectures.

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

## LLM USAGE

We employed LLMs as a general-purpose assistive tool of this work. Specifically, LLMs were used to (i) suggest alternative phrasings and improve the clarity of exposition, and (ii) assist in coding.

---

**Algorithm 1** ReasonBreak Adversarial Image Generation

---

1: **Input:** Image $I$, geographic concepts $\boldsymbol{c}$ and bounding boxes $\boldsymbol{g}$ for $I$, trained decoder $\mathcal{G}_\theta$, pre-computed embedding bank $\mathcal{E}$, pre-trained text encoder $\psi_t$, perturbation budget $\epsilon$, max blocks $N_{\max}$.
2: **Output:** Adversarial image $I'$.
3: **procedure** REASONBREAK-GENERATE($I, \boldsymbol{c}, \boldsymbol{g}, \mathcal{G}_\theta, \mathcal{E}, \psi_t, \epsilon, N_{\max}$)
4:     $\{B_k\}_{k=1}^N \leftarrow$ AdaptiveDecomposition($I, N_{\max}$)           ▷ Equation 3
5:     $\{\mathcal{C}_k\}_{k=1}^N \leftarrow$ AssignConcepts($\{B_k\}, \boldsymbol{c}, \boldsymbol{g}$)
6:     $\{\hat{B}_k\}_{k=1}^N \leftarrow$ empty list           ▷ To store perturbed blocks
7:     **for** each block $B_k$ and concept set $\mathcal{C}_k$ **do**
8:         $\mathbf{e}_{\text{prior}}^k \leftarrow \arg\min_{\mathbf{e}\in\mathcal{E}} \max_{c\in\mathcal{C}_k} \cos(\psi_t(c), \mathbf{e})$           ▷ Equation 4
9:         $\delta_k \leftarrow \mathcal{G}_\theta(\mathbf{e}_{\text{prior}}^k)$
10:         $B_k' \leftarrow B_k + \delta_k$
11:         $\hat{B}_k \leftarrow$ clip($B_k', B_k - \epsilon, B_k + \epsilon$)           ▷ Enforce $L_\infty$ constraint
12:         Append $\hat{B}_k$ to $\{\hat{B}_k\}$
13:     **end for**
14:     $I' \leftarrow$ ReconstructImage($\{\hat{B}_k\}$)
15:     **return** $I'$
16: **end procedure**

---

## A    DECODER ARCHITECTURE

The architecture of our learnable decoder $\mathcal{G}_\theta$, which translates a conceptual prior embedding into an adversarial perturbation, is detailed in Algorithm 2. The decoder is primarily composed of a series of residual blocks (`ResBlock`) and upsampling blocks (`UpBlock`), as specified in Algorithms 3 and 4.

---

**Algorithm 2** Decoder Architecture ($\mathcal{G}_\theta$)

---

**Require:** Input embedding $\mathbf{e} \in \mathbb{R}^{B \times D}$, where $B$ is batch size and where $D$ is embedding size
**Require:** Target image size $H, W$, and target channels $C$
**Ensure:** Adversarial perturbation $\delta \in \mathbb{R}^{B \times C \times H \times W}$
1: $h_{\text{init}} \leftarrow H/16$
2: $x \leftarrow$ Linear($\mathbf{e}$)
3: $x \leftarrow$ Reshape($x, (B, 256, h_{\text{init}}, h_{\text{init}})$)
4: $x \leftarrow$ ResBlock($x, \text{in\_ch} = 256, \text{out\_ch} = 256$)
5: $x \leftarrow$ UpBlock($x, \text{in\_ch} = 256, \text{out\_ch} = 128$)
6: $x \leftarrow$ ResBlock($x, \text{in\_ch} = 128, \text{out\_ch} = 128$)
7: $x \leftarrow$ UpBlock($x, \text{in\_ch} = 128, \text{out\_ch} = 64$)
8: $x \leftarrow$ ResBlock($x, \text{in\_ch} = 64, \text{out\_ch} = 64$)
9: $x \leftarrow$ UpBlock($x, \text{in\_ch} = 64, \text{out\_ch} = 32$)
10: $x \leftarrow$ ResBlock($x, \text{in\_ch} = 32, \text{out\_ch} = 32$)
11: $x \leftarrow$ UpBlock($x, \text{in\_ch} = 32, \text{out\_ch} = 16$)
12: $x \leftarrow$ ResBlock($x, \text{in\_ch} = 16, \text{out\_ch} = 16$)
13: $x \leftarrow$ Conv2d($x, \text{in\_ch} = 16, \text{out\_ch} = C, \text{kernel} = 3, \text{padding} = 1$)
14: $\delta \leftarrow$ Tanh($x$)
15: **return** $\delta$

---

---

**Algorithm 3** ResBlock Module

---

1: **procedure** RESBLOCK($x$, in_ch, out_ch)
2:     $r \leftarrow$ Conv2d($x$, in_ch, out_ch, kernel $= 1$)
3:     $h \leftarrow$ Conv2d($x$, in_ch, out_ch, kernel $= 3$, padding $= 1$)
4:     $h \leftarrow$ BatchNorm2d($h$)
5:     $h \leftarrow$ LeakyReLU($h, \alpha = 0.2$)
6:     $h \leftarrow$ Conv2d($h$, out_ch, out_ch, kernel $= 3$, padding $= 1$)
7:     $h \leftarrow$ BatchNorm2d($h$)
8:     $h \leftarrow$ EfficientAttention($h$)
9:     $h \leftarrow h + r$
10:    $h \leftarrow$ LeakyReLU($h, \alpha = 0.2$)
11:    **return** $h$
12: **end procedure**

---

**Algorithm 4** UpBlock Module

---

1: **procedure** UPBLOCK($x$, in_ch, out_ch)
2:     $h \leftarrow$ Upsample($x$, scale_factor $= 2$, mode='nearest')
3:     $h \leftarrow$ Conv2d($h$, in_ch, out_ch, kernel $= 3$, padding $= 1$)
4:     $h \leftarrow$ BatchNorm2d($h$)
5:     $h \leftarrow$ LeakyReLU($h, \alpha = 0.2$)
6:    **return** $h$
7: **end procedure**

---

# B  DATASET CONSTRUCTION DETAILS

## B.1  THREE-STAGE ANNOTATION PIPELINE

The construction of GeoPrivacy-6K employs a systematic three-stage annotation pipeline implemented using QwenVL 2.5 72B as the annotation model. To mitigate potential factual inaccuracies from model limitations, our annotation process focuses exclusively on visual feature characterization rather than specific geographic location identification. This multi-stage approach progressively refines image content from basic geographic filtering to detailed hierarchical concept analysis and precise spatial reasoning chain extraction, ensuring comprehensive capture of the visual-conceptual relationships that MLRMs exploit during geographic inference while maintaining annotation quality and consistency.

### B.1.1  STAGE 1: GEOGRAPHIC CONTENT FILTERING

The initial filtering stage identifies images containing real-world geographical features suitable for location inference training. This stage operates through automated resolution screening followed by content-based evaluation that excludes abstract patterns, studio portraits with plain backgrounds, or isolated object close-ups while retaining images with identifiable natural landmarks, architectural elements, or environmental characteristics.

> **Stage 1 Prompt:** The system evaluates whether images contain real-world geographical features (natural or man-made elements related to places on Earth) while excluding abstract patterns, studio portraits, or isolated object close-ups. The assessment produces a boolean decision with reasoning explanation in JSON format.

### B.1.2  STAGE 2: HIERARCHICAL SCENE ANNOTATION

Images passing the geographic filter undergo comprehensive hierarchical categorization that captures the conceptual structure employed by MLRMs during visual analysis. This stage establishes the foundational semantic framework through three-level hierarchical classification and detailed attribute annotation across environmental, architectural, and atmospheric dimensions.

The hierarchical framework begins with **L1 - Environmental Domain** classification, distinguishing between Natural Environment and Built Environment contexts. This guides subsequent **L2 - Contextual Setting** refinement, where natural environments are classified into mountainous, forest/woodland, plains/grassland, water body, desert, or coastal categories, while built environments encompass urban/city, rural/suburban, transportation infrastructure, or industrial settings. The **L3 - Scene Specification** level provides granular scene categorization, subdividing urban environments into street views, skylines, plazas/parks, residential areas, commercial districts, or historic districts, while mountainous regions distinguish between peaks/ridges, valleys, or plateaus.

Beyond hierarchical scene classification, the annotation framework captures detailed descriptive attributes including environmental elements (both natural features such as vegetation, trees, rock formations, water bodies, and man-made elements including buildings, roads, vehicles, infrastructure), architectural characteristics (styles ranging from modern to classical/historic, and construction materials from brick/stone to glass curtain walls), and atmospheric conditions (temporal factors like lighting, weather and environmental characteristics).

> **Stage 2 Prompt:** The system categorizes images using a three-level hierarchy (L1: Environmental Domain, L2: Contextual Setting, L3: Scene Specification) while capturing detailed descriptive attributes across environmental elements (natural and man-made), architectural characteristics (styles and materials), and atmospheric conditions (lighting, weather).

### B.1.3 STAGE 3: GEOGRAPHIC REASONING CHAIN EXTRACTION

The final and most critical stage generates the hierarchical reasoning chains that mirror MLRM geographic inference processes. This stage produces the concept-region mappings essential for training ReasonBreak by systematically analyzing visual evidence through four geographic scales: continental, national, city, and local levels. Each reasoning step identifies a specific visual concept and its precise spatial location through normalized square bounding boxes.

> **Stage 3 Prompt:** The system performs hierarchical geographic reasoning analysis (Continental → National → City → Local) identifying key visual concepts at each level with precise spatial localization. Each reasoning step produces descriptive concept phrases (5-10 words) with normalized square bounding boxes [center_x, center_y, size] and confidence scores, generating the concept-region mappings essential for adversarial training.

### B.2 DATA COLLECTION AND SOURCE INTEGRATION

Our data collection process sources high-quality images from three established computer vision datasets that provide complementary geographic coverage. HoliCity (Zhou et al., 2020) contributes diverse urban scenes with detailed architectural elements and city landscapes, Aesthetic-4K (Zhang et al., 2025b) provides visually compelling natural and built environments with strong compositional quality, and LHQ (Skorokhodov et al., 2021) offers ultra-high-resolution landscape images spanning diverse geographical regions and environmental conditions.

The technical filtering process ensures all images maintain a minimum resolution of 2048 pixels along at least one dimension. Subsequently, the three-stage annotation pipeline transforms raw images into a comprehensive dataset with hierarchical scene categorization, detailed attribute annotation, and precise concept-region mappings through geographic reasoning chain extraction.

### B.3 INFERENCE DIFFICULTY ASSESSMENT

Inference difficulty ratings are determined based on confidence scores generated during the geographic reasoning analysis stage. Easy cases (17.8%) feature obvious, globally distinctive landmarks or features that enable straightforward location inference. Medium difficulty cases (29.1%) require regional-level geographic knowledge and more sophisticated visual analysis. Hard cases (53.2%) demand fine-grained local geographic reasoning and represent the most challenging scenarios for both human experts and automated systems.

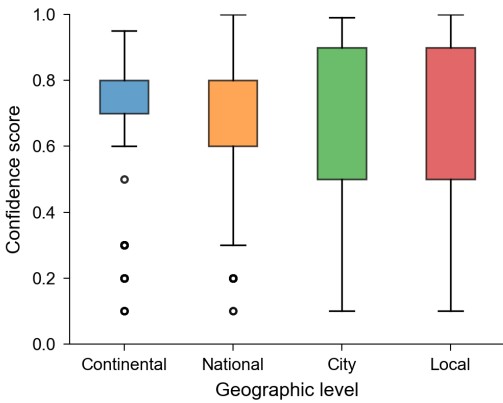

Figure 7: Distribution of confidence scores across geographic inference levels. Higher confidence scores indicate greater certainty in geographic predictions. The predominance of high confidence scores at the city and local levels demonstrates the sophisticated reasoning capabilities required for precise location inference.

Figure 7 illustrates the confidence score distribution across different geographic inference levels, demonstrating the challenging nature of our dataset composition. The prevalence of high-confidence scores at city and local levels reflects the sophisticated reasoning capabilities required for precise location inference and validates the complexity of our curated dataset.

This comprehensive three-stage annotation structure enables precise concept-region mapping essential for training ReasonBreak's concept-aware adversarial generator, providing the granular supervision necessary for targeted perturbation generation across diverse geographic inference scenarios while maintaining the spatial precision required for effective reasoning pathway disruption.

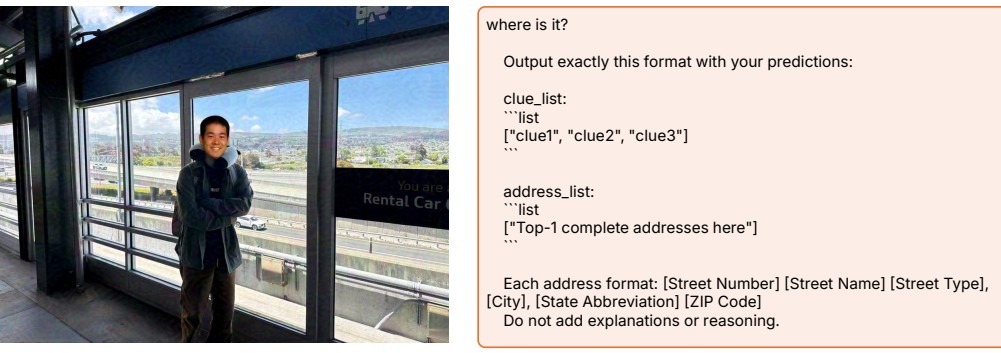

Figure 8: Demonstration of input sensitivity in MLRMs. Adding a single line break to the prompt causes InternVL 3.0 72B to generate drastically different location inferences.

## C    COUNTER-INTUITIVE SCALING PHENOMENA IN REASONING MODELS

Our experiments reveal two intriguing phenomena rarely observed in traditional perception models but consistently present in MLRMs, particularly in open-source models like InternVL 3.0 72B. First, an Inverted Scaling Relationship: unlike traditional adversarial attacks where larger perturbations typically yield stronger effects, we observe instances in MLRMs where smaller perturbations occasionally produce more effective attacks. Second, the Adversarial Enhancement Effect: while adversarial noise typically degrades model performance in traditional perception models, we occasionally observe anomalous cases in MLRMs where adversarial perturbations actually improve

model performance, resulting in negative protection rates. In Table 1, we normalize these occasional negative values to zero while discussing this phenomenon separately here.

We attribute these phenomena to two key factors: First, the inherent randomness introduced by the LLM component in MLRMs. For instance, model temperature settings introduce inherent stochasticity in outputs, making some performance variations expected. More surprisingly, the second factor relates to input sensitivity in reasoning models. Figure 8 demonstrates this phenomenon: on InternVL 3.0 72B, even with temperature=0, simply adding a line break at the end of the prompt transforms the output from *[Rental Car sign", Highway view", Urban landscape"], address list: [100 Rental Car Center, San Francisco, CA 94130"]"* to *[Rental Car", highway view", train station"], address list: [1000 Broadway, Oakland, CA 94607"]"*. Similarly, this sensitivity extends to image inputs, where ostensibly adversarial perturbations can occasionally trigger patterns that improve model accuracy.

These observations highlight the complex nature of the reasoning processes of MLRMs. Understanding and addressing these unique characteristics presents an important direction for future research in privacy protection against reasoning-based models.

## D  VISUAL QUALITY ANALYSIS

We provide qualitative analysis of the visual quality of adversarial examples generated by Reason-Break and baseline methods across different perturbation budgets. Figure 9 presents representative examples of adversarial images generated under $\epsilon = 8/255$ and $\epsilon = 16/255$ constraints. All methods produce perturbations that remain largely imperceptible to human observers, ensuring that privacy protection does not compromise image usability for legitimate sharing purposes. While the overall visual impact is minimal across all methods, we observe distinct perturbation patterns. Baseline methods (AnyAttack, M-Attack) exhibit subtle block-like artifacts, particularly noticeable in high-resolution images. This occurs because these methods generate perturbations at lower resolutions and resize them to match the target image dimensions, leading to slight pixelation effects. In contrast, our concept-aware approach produces more naturally distributed perturbations that align with semantic boundaries and geographic features, avoiding the block artifacts inherent in resize-based approaches.

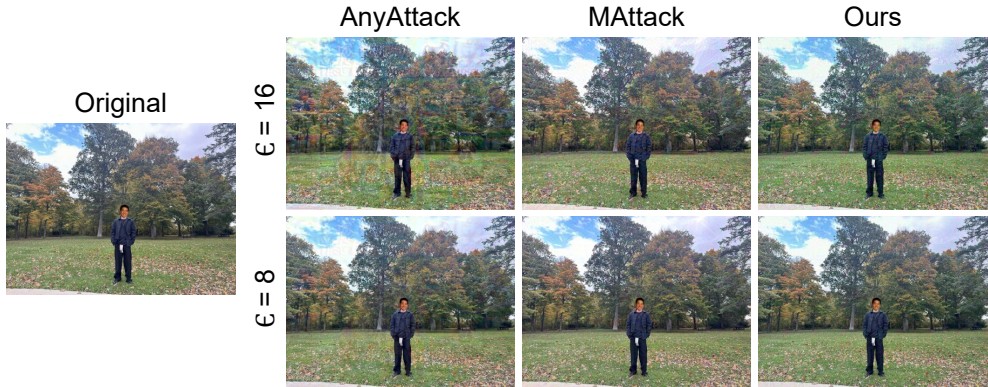

Figure 9: Visual comparison of adversarial examples generated by different methods.

## E  QUALITATIVE EXAMPLES OF GEOPRIVACY-6K

To facilitate a deeper understanding of the GeoPrivacy-6K dataset and validate the effectiveness of our automated annotation pipeline, we present representative visualizations in Figure 10. These examples demonstrate the diversity of scenes covered, ranging from dense urban environments to remote natural landscapes. As illustrated, the annotations generated by QwenVL 2.5 72B follow a structured geographic reasoning chain. The process initiates with broad environmental classification (e.g., "European-style urban infrastructure") and progressively narrows down to localized,

discriminative features (e.g., specific road markings or distinct mountain peaks). Crucially, each reasoning step is grounded by a normalized square bounding box parameterized as `[center_x, center_y, size]` alongside a confidence score.

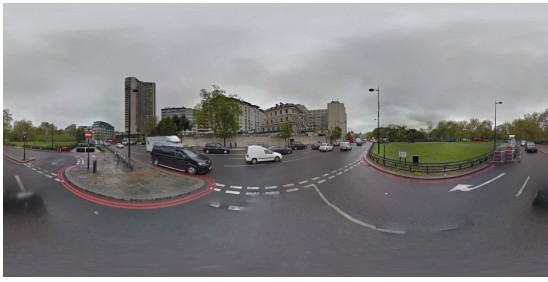

step_1 | key_concept: "European-style urban infrastructure" | square_bbox: [0.5,0.5,1.0] | confidence: 0.9

step_2 | key_concept: "Red double-decker bus (UK icon)" | square_bbox: [0.7,0.45,0.1] | confidence: 0.95

step_3 | key_concept: "London-specific road markings" | square_bbox: [0.5,0.6,0.2] | confidence: 0.9

step_4 | key_concept: "'Achilles Way' street sign" | square_bbox: [0.75,0.5,0.05] | confidence: 0.95

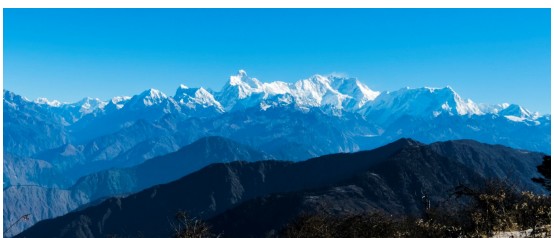

step_1 | key_concept: "High-altitude snow-capped mountains" | square_bbox: [0.5,0.3,0.2] | confidence: 0.95

step_2 | key_concept: "Prominent Himalayan peak with sharp ridges" | square_bbox: [0.45,0.3,0.1] | confidence: 0.9

step_3 | key_concept: "Himalayan foothills with clear visibility" | square_bbox: [0.5,0.6,0.2] | confidence: 0.85

step_4 | key_concept: "Central peak resembling Kangchenjunga" | square_bbox: [0.45,0.3,0.1] | confidence: 0.8

Figure 10: Visualization of hierarchical annotations in GeoPrivacy-6K. The figure displays two samples with their corresponding automated reasoning chains.

Table 3: Privacy protection rates under different JPEG compression quality factors ($Q$) on InternVL 3.0 72B. The method demonstrates strong stability even under aggressive compression ($Q = 50$).

| Quality Factor | Region | Metro. | Tract | Block |
|---|---|---|---|---|
| $Q = 95$ (Default) | 10.8 | 0.0 | 33.3 | 58.3 |
| $Q = 75$ | 11.1 | 0.0 | 33.3 | 58.3 |
| $Q = 50$ | 9.3 | 0.0 | 30.0 | 58.3 |

## F  COMPUTATIONAL EFFICIENCY ANALYSIS

We evaluate the computational efficiency of ReasonBreak against the baseline methods, focusing on both training overhead and inference latency. Regarding training costs, there are substantial disparities among approaches. The generator-based baseline, AnyAttack, requires a computationally intensive pre-training phase spanning approximately one week on three NVIDIA A100 GPUs. In contrast, ReasonBreak significantly reduces this overhead, converging in 6 hours and 30 minutes on a single GPU. The PGD-style baseline, M-Attack, incurs no training cost as it computes perturbations dynamically at inference time.

For inference, we measured the time required to generate adversarial examples for DoxBench ($\approx$500 images). M-Attack exhibits the highest latency (43 minutes and 30 seconds) due to the necessity of iterative gradient optimization for each input. Generator-based methods demonstrate a marked advantage in deployment efficiency: AnyAttack completes the process in 2 minutes and 30 seconds, while ReasonBreak requires 5 minutes and 20 seconds. The marginal increase in our inference time compared to AnyAttack is attributable to the adaptive decomposition and concept assignment pre-processing steps. This indicates that ReasonBreak achieves a favorable balance, offering protection rates comparable to computationally expensive methods while maintaining the near real-time inference capabilities of generator-based architectures.

## G    ROBUSTNESS TO JPEG COMPRESSION

To verify the practicality of ReasonBreak in real-world social media environments, where uploaded images typically undergo lossy compression, we evaluated the resilience of our generated perturbations against varying levels of JPEG compression. It is important to note that all experimental results reported in the main text were conducted using a standard JPEG quality factor ($Q$) of 95 to simulate a realistic baseline. In this section, we perform a stress test by further reducing the quality factor to $Q = 75$ and $Q = 50$. We utilize InternVL 3.0 72B as the target model for this evaluation.

As shown in Table 3, ReasonBreak exhibits remarkable stability. Reducing the quality factor from 95 to 75 results in virtually no degradation in protection performance. Even under aggressive compression ($Q = 50$), the decline in protection rates is minimal. This resilience suggests that the concept-aware perturbations generated by our method are structurally robust and can survive the standard image processing pipelines employed by major social platforms.

