# OpenReview forum: "Disrupting Hierarchical Reasoning: Adversarial Protection for Geographic Privacy in Multimodal Reasoning Models"
_ICLR.cc/2026/Conference — ICLR 2026 Poster_

### Official Review · Reviewer_kHCB · 2025-10-20

**Soundness:** 3
**Presentation:** 3
**Contribution:** 3
**Rating:** 6
**Confidence:** 3

**Summary:**

This paper proposes a defense method that aims to disrupt geographic reasoning in large multimodal reasoning models by adding perturbation noise. The authors conduct experiments across multiple models and release the GeoPrivacy-6K dataset, which contains high-resolution, spatially localized images annotated with bounding boxes. The study addresses an important privacy issue and contributes a new dataset and adversarial defense framework.

**Strengths:**

The work targets an emerging privacy threat related to geographic inference from images by large reasoning models.

The authors propose an adversarial approach that effectively disrupts hierarchical reasoning through concept-aware perturbations.

The new GeoPrivacy-6K dataset is a valuable contribution, containing ultra-high-resolution and spatially localized images.

**Weaknesses:**

Dataset Size and Analysis. It would strengthen the paper to analyze which types of images are more vulnerable to attacks and which are more resistant. Please summarize the visual or semantic characteristics of easily attacked versus robust images.

Computational Performance. Include a comparison of running time and GPU computation cost between ReasonBreak and other defense baselines.
Resolution Impact. The paper uses ultra-high-resolution images. Please explain why this resolution level was chosen and analyze whether image resolution affects attack or defense effectiveness.

Dataset Relationships. Clarify the relationship between GeoPrivacy-6K and DOXBENCH. Are they complementary or overlapping?

Defense Comparison. Compare the performance of your defense method (ReasonBreak) against differential privacy (DP) methods implemented in DOXBENCH or similar benchmarks.

Appendix Suggestion. It would be helpful to include sample images from GeoPrivacy-6K in the appendix to illustrate the dataset’s diversity, annotation quality, and spatial localization.

**Questions:**

see Weakness

---

> ### Author Response · Authors · 2025-11-20
>
> **Part 1**
>
> We thank you for your positive assessment and for specifically highlighting the value of our GeoPrivacy-6K dataset and the effectiveness of the concept-aware adversarial framework. We are encouraged that you find our work addresses an important emerging privacy threat.
>
> We have taken your constructive suggestions for deeper experimental analysis seriously. In the revised manuscript, we have incorporated a Computational Efficiency Analysis (Appendix C), a Failure Case Analysis (Section 5.5), and added visual samples of the dataset (Appendix B.2) to address your specific requests. Revisions in the PDF are highlighted in blue for clarity. Below, we provide detailed responses to each point.
>
> ---
>
> ### 1. Dataset Size and Vulnerability Analysis
>
> > **Comment:** *"Dataset Size and Analysis. It would strengthen the paper to analyze which types of images are more vulnerable to attacks and which are more resistant..."*
>
> **Response:**
> We appreciate your interest in the dataset characteristics. The dataset size and composition were detailed in Section 3 and Appendix B.
>
> Regarding the vulnerability analysis: We found your suggestion to investigate which images are "vulnerable" or "resistant" to be highly insightful. It motivated us to conduct a dedicated Qualitative Failure Analysis (Section 5.5), which yielded fascinating results regarding the boundary of adversarial protection.
>
> We found that our method faces challenges when images contain explicit, high-saliency text (e.g., specific street names or logos like "1565, B46, Google"). In these cases, the MLRM shifts its modality from visual reasoning to Optical Character Recognition (OCR), bypassing the concept-based features we target. This analysis provides a valuable nuance to our contribution, clarifying that while we effectively secure visual reasoning, textual identification remains a distinct challenge.
>
> ### 2. Computational Performance
>
> > **Comment:** *"Include a comparison of running time and GPU computation cost between ReasonBreak and other defense baselines."*
>
> **Response:**
> We thank you for this practical suggestion. We have added a detailed comparison in **Appendix F** of the revised manuscript. The table below summarizes the costs:
>
> | Method | Training Time | Training Resources | Inference Time (Total) |
> | :--- | :---: | :---: | :---: |
> | M-Attack | N/A | N/A | ~43 min 30s |
> | AnyAttack | ~1 week | 3 $\times$ A100 GPUs | ~2 min 30s |
> | ReasonBreak (Ours) | ~6.5 hours | 1 $\times$ A800 GPU | ~5 min 20s |
>
> * Training: ReasonBreak is drastically more efficient than the baseline AnyAttack, reducing training time from weeks to hours while using only a single GPU.
> * Inference: Unlike optimization-based methods (M-Attack) which require heavy computation per image, our method achieves near real-time performance.
> * Conclusion: By combining extremely low training costs with fast inference, ReasonBreak offers a highly practical solution for real-world deployment.
>
> ### 3. Resolution Impact
>
> > **Comment:** *"The paper uses ultra-high-resolution images. Please explain why this resolution level was chosen and analyze whether image resolution affects attack or defense effectiveness."*
>
> **Response:**
> The choice of high-resolution images is driven by the specific nature of the MLRM threat. Modern social media platforms typically standardize images to kilopixel-level resolutions (e.g., 1080px for Instagram, and 2048px for Facebook/Meta), which preserve the fine-grained details—such as distant signage or architectural textures—that MLRMs require for precise geolocation. In contrast, lower-resolution images naturally lose these critical details, making them inherently less vulnerable to reasoning-based attacks simply because the visual evidence is absent. Therefore, our work specifically targets the high-risk scenario of high-quality uploads where privacy leakage is a genuine concern.
>
> To analyze how quality degradation affects defense effectiveness, we conducted a stress test under varying levels of compression in the **new Appendix G**. Our results demonstrate that ReasonBreak remains highly effective (maintaining a 58.3% Block-level protection rate) even under aggressive quality reduction ($Q=50$), confirming that our method is robust to the resolution and quality variations typical of real-world platforms.

---

> > ### Comment · Reviewer_kHCB · 2025-11-26
> > **Response to Author's Rebuttal**
> >
> > I appreciate the authors' detailed response. All of my concerns have been addressed. I have decided to raise my score. However, I recommend using the same GPU(A100 or A800) to show the running time of different methods.

---

> ### Author Response · Authors · 2025-11-20
>
> **Part 2**
>
> ### 4. Dataset Relationships (GeoPrivacy-6K vs. DoxBench)
>
> > **Comment:** *"Clarify the relationship between GeoPrivacy-6K and DOXBENCH. Are they complementary or overlapping?"*
>
> **Response:**
> We clarify that the two datasets are completely disjoint and complementary, serving distinct roles in our framework:
>
> * GeoPrivacy-6K (Our Contribution): This serves as the Training Set. It consists of 6,341 high-resolution images annotated with hierarchical concepts but without using their GPS coordinates for training. It is used solely to train the ReasonBreak generator to learn generalized concept-aware perturbations.
> * DoxBench (Existing Benchmark): This serves as the Testing Set. It contains 500 distinct images with ground-truth GPS coordinates. We use it strictly for evaluating the privacy protection rate of our method.
> * Independence: There is zero overlap between the two datasets. This setup ensures that our evaluation tests the true generalization capability of the defense on unseen images, avoiding any data leakage.
>
> ### 5. Defense Comparison (vs. Differential Privacy)
>
> > **Comment:** *"Compare the performance of your defense method (ReasonBreak) against differential privacy (DP) methods implemented in DOXBENCH or similar benchmarks."*
>
> **Response:**
> We appreciate the reviewer raising the comparison with Differential Privacy (DP). We respectfully clarify that we did not include standard DP methods (typically implemented via Gaussian or Laplacian noise injection) as baselines because they are fundamentally ill-suited for this specific threat model due to the Privacy-Utility Trade-off.
>
> * Incompatibility with Visual Utility: DP mechanisms provide statistical guarantees by adding random noise. However, to effectively disrupt the fine-grained reasoning of robust MLRMs (e.g., obscuring a specific street sign or architectural texture), the magnitude of this random noise must be substantial. Such high-variance noise inevitably degrades the image quality to a point where it becomes visually unpleasant or unrecognizable, violating the core requirement of social media sharing.
> * Why We Chose Adversarial Baselines: In contrast, ReasonBreak targets specific semantic concepts through optimized perturbations under a strict imperceptibility constraint (Equation 5). This allows us to achieve protection without destroying visual utility. Consequently, we focused our comparison on state-of-the-art adversarial attacks (AnyAttack, M-Attack), which represent the practical standard for utility-preserving privacy protection.
>
> ### 6. Appendix Suggestion (Sample Images)
>
> > **Comment:** *"It would be helpful to include sample images from GeoPrivacy-6K in the appendix to illustrate the dataset’s diversity, annotation quality, and spatial localization."*
>
> **Response:**
> We appreciate your interest in the dataset details.
>
> First, we wish to highlight that the comprehensive specifications regarding dataset statistics, annotation protocols, and hierarchical definitions were already documented in Appendix B (Dataset Construction Details) of the original submission.
>
> Second, we sincerely thank you for the valuable suggestion to visualize these data samples. In response, we have explicitly added a new section, "Qualitative Examples of GeoPrivacy-6K Annotations," along with Figure 10 in the revised Appendix. This visualization showcases representative samples to demonstrate the diversity of scenes, the quality of the automated hierarchical reasoning chains, and the bounding boxes.
>
> ---
>
> We hope these additional analyses and visual examples fully address your comments.

---

### Official Review · Reviewer_6ECh · 2025-10-22

**Soundness:** 4
**Presentation:** 2
**Contribution:** 3
**Rating:** 4
**Confidence:** 3

**Summary:**

This paper proposes a novel attack approach to break the mulitmodal large reasoning model (MLRM) for performing geographical inference. To support this, the authors introduce a new dataset called GeoPrivacy-6K, which contains 6,000 more images and concept annotations for training. Experiments on DOXBENCH demonstrate that the proposed method achieves superior results compared to previous approaches on both open- and closed-source models.

**Strengths:**

* The motivation of the paper is intriguing and highly relevant, especially for protecting users’ geographical information in line with existing privacy regulations.
* The proposed idea is supported by reasonable theoretical analysis, making the design both robust and practical.
* The experiments are thorough, and the results are impressive. In particular, the introduction of the protected success rate metric, instead of the usual attack success rate, is a valuable addition for future evaluations.

**Weaknesses:**

* While the paper is well-motivated, the methodology is somewhat difficult to follow. For example, in Equation (3), the meaning of defining $(m^\*, n^\*)$ is unclear. Do the authors mean that all $(m^\*, n^\*)$ follow the stated condition? Also, terms like "concept subset" and "spatial overlap analysis" are mentioned without sufficient explanation. It would help to clearly define these terms or include a figure illustrating the training pipeline.
* The proposed min-max target selection involves finding a "null point" from the database. This might be a limitation because if an image is dissimilar or unrelated to the database, the results could be negatively affected. Moreover, the reasoning behind selecting $\min \{ \mathbf{e} \in \mathcal{E} \}$ is not fully explained and would benefit from further clarification.
* The choice of $N_{\text{max}}$ is not entirely convincing. As shown in Figure 4, when $N_{\text{max}} > 16$, the PRR of the "metro" class drops to zero. Although other classes improve, this choice does not appear optimal. It would be helpful to explain why the "metro" class is considered less important or why this trade-off is acceptable.

---

**Overall**: The paper addresses an important and timely topic with clear motivation and solid contributions. The results are promising and provide meaningful insights for the community. However, the methodological descriptions are sometimes unclear and would benefit from additional clarification. My current score is 4, and I would be happy to raise it once these issues are addressed.

**Questions:**

The questions are mainly about the weaknesses.

1.	What are the "concepts" used for training?
2.	Does using the database as an anchor point affect the results for images that fall outside the database distribution?
3.	Is the inclusion of $\min \{ \mathbf{e} \in \mathcal{E} \}$ intended to prevent the reconstructed error from deviating too far from the original images?
4.	Could the authors elaborate further on the selection of $N_{\text{max}}$ and its effect on different classes?

---

> ### Author Response · Authors · 2025-11-20
>
> **Part 1**
>
> We thank you for the detailed evaluation and for recognizing the "excellent soundness" and "robust design" of our proposed approach. We are heartened that your assessment of the core contributions is highly positive, and we believe the concerns raised regarding the methodology stem primarily from the clarity of our initial presentation, rather than fundamental limitations.
>
> We have taken your feedback regarding presentation clarity very seriously. In the revised manuscript, we have extensively rewritten **Section 4.3** and added a new **Figure 3** to visually illustrate the entire training pipeline. Revisions in the PDF are highlighted in blue for clarity. Below, we address your specific methodological questions and clarify the design choices.
>
> ---
>
> ### 1. Methodological Clarity (Eq. 3, Terminology, and Pipeline Figure)
>
> > **Comment:** *"Methodology is somewhat difficult to follow... Eq (3) meaning... terms like 'concept subset' and 'spatial overlap analysis'... include a figure illustrating the training pipeline."*
>
> **Response:**
> We appreciate your attention to the formal definitions.
>
> **Regarding Equation (3):** We wish to clarify the nature of the $\arg\min$ operator used here. It does not imply that all pairs $(m, n)$ satisfy the condition; rather, it denotes an optimization operation that searches through valid candidates to select the single optimal pair $(m^\ast, n^\ast)$ that minimizes the objective function.
>
> * **Concrete Example:** Consider an input image with $W=1000, H=1000$ and $N_{\text{max}}=64$. The operator evaluates all integer pairs where $mn \le 64$. It will uniquely identify and return $m^\ast=8, n^\ast=8$, as this specific pair yields a perfect aspect ratio match (Error = 0) while maximizing resolution. This ensures our adaptive decomposition preserves geometric structure without distortion.
>
> **Regarding the Pipeline and Terminology:** We agree that visual aids are essential. In the revised manuscript, we have added a comprehensive **Figure 3** to visually illustrate the training pipeline. As summarized in the figure's caption, the process follows five strict steps:
>
> 1.  The input image undergoes Adaptive Decomposition into an $m^\ast \times n^\ast$ grid of blocks (calculated via Eq. 3).
> 2.  Each block $B_k$ is assigned a set of relevant concepts $\mathcal{C}_k$ via spatial overlap analysis. As shown in the dashed boxes of Figure 3, this results in three outcomes: a single concept, multiple concepts, or the default set (all concepts) if no spatial overlap exists.
> 3.  The Minimax Target Selection uses the relevant concepts and the Embedding Bank $\mathcal{E}$ to find a hard-negative prior $\mathbf{e}_{\text{prior}}^{k}$.
> 4.  This prior is fed into the learnable Decoder $\mathcal{G}_{\theta}$ to synthesize a block-specific perturbation $\delta_k$.
> 5.  The final adversarial image $I'$ is reconstructed by adding perturbations to their corresponding clean blocks.

---

> ### Author Response · Authors · 2025-11-20
>
> **Part 2**
>
> ### 2. Minimax Target Selection and Embedding Bank
>
> > **Comment:** *"The proposed min-max target selection involves finding a 'null point' from the database. This might be a limitation because if an image is dissimilar or unrelated to the database, the results could be negatively affected."*
>
> **Response:**
> We appreciate the opportunity to clarify the underlying mechanism of our target selection. It is crucial to emphasize that our Minimax strategy operates on the principle of **Repulsion (Hard-Negative Mining)**, rather than Retrieval or Matching. The concern that results might be negatively affected if an image is "dissimilar or unrelated to the database" implies a premise where the system seeks semantic alignment. However, our objective is the exact opposite: we search for a target $\mathbf{e}_{\text{prior}}^k$ that minimizes cosine similarity to the input block (i.e., maximizes semantic distance). In this context, having concepts in the database that are semantically distant from the input is not a limitation, but a functional requirement for generating effective adversarial gradients.
>
> To illustrate this with a concrete example, consider an input block depicting a "Desert". If our Embedding Bank contained only semantically unrelated concepts, such as "Rainforests", this would actually represent an ideal scenario for our attack. The algorithm would identify the "Rainforest" embedding as the optimal hard-negative prior ($\mathbf{e}_{\text{prior}}^k$) precisely *because* it is maximally distant from the "Desert" concept. The generator is then conditioned on this prior to synthesize perturbations that shift the image representation away from the true "Desert" manifold and towards this "Rainforest" conceptual void, effectively breaking the reasoning chain.
>
> Therefore, the Embedding Bank $\mathcal{E}$ functions as a diverse vocabulary of "incorrect answers" rather than a matching gallery. As long as the database contains concepts distinct from the input—which is statistically guaranteed given the diversity of the GeoPrivacy-6K dataset—it provides valid repulsive anchors to drive the perturbation process regardless of whether the input image has a "similar" counterpart in the bank.
>
> ---
>
> ### 3. Hyperparameter Choice ($N_{\text{max}}=64$) and Trade-offs
>
> > **Comment:** *"The choice of N_max=64 is not entirely convincing. As shown in Figure 4, when N_max=64, the PRR of the 'metro' class drops to zero... It would be helpful to explain why the 'metro' class is considered less important or why this trade-off is acceptable."*
>
> **Response:**
> We thank you for highlighting this trade-off. To fully justify this design choice, it is essential to revisit the definitions of the geographic hierarchy as outlined in Section 5.1 (based on standard US Census Bureau designations):
>
> * **Metro. (Metropolitan Statistical Area):** Represents a vast region (e.g., Greater New York), typically covering 2,000–10,000 $\text{km}^2$.
> * **Tract & Block:** Represent specific neighborhoods and street segments, covering only 1–5 $\text{km}^2$ and 0.005–0.05 $\text{km}^2$, respectively.
>
> **Justification for the Trade-off:**
> The choice of $N_{\text{max}}=64$ reflects a strategic prioritization of **anti-doxing** over general obfuscation.
>
> 1. The privacy risk is highly asymmetric. Identifying that a user is in the "Greater New York" (Metro.) is often low-risk or even public knowledge. However, pinpointing their exact neighborhood (Tract) or street address (Block) constitutes a severe privacy violation ("doxing").
> 2.  As shown in the ablation, $N_{\text{max}}=64$ yields the highest protection rates for the critical **Tract and Block** levels. We explicitly accept the performance drop at the coarse Metro. level to maximize protection where it matters most: preventing the precise localization of a user's home or current street.

---

> ### Author Response · Authors · 2025-11-20
>
> **Part 3**
>
> ### 4. Clarification on "Concepts" Used for Training
>
> > **Question:** *"What are the 'concepts' used for training?"*
>
> **Response:**
> We appreciate the opportunity to clarify the semantic granularity of our training data.
>
> Intuitively: We refer the reviewer to Figure 1 of our original submission, which explicitly visualizes the hierarchy of concepts used in our framework. As shown in that figure, these concepts range from broad environmental indicators (e.g., the Continental-level concept "English language high-end retailer storefronts") to distinctive local landmarks (e.g., the City-level concept "Distinctive clock facade on Tiffany & Co. building") and fine-grained details (e.g., the Local-level concept "Large circular fountain").
>
> Formally: These concepts are generated via a rigorous three-stage annotation pipeline (Environmental Domain $\to$ Contextual Setting $\to$ Scene Specification). For the complete taxonomy and generation protocol, we kindly refer the reviewer to Appendix B (Dataset Construction Details), where the prompt templates and hierarchical definitions are comprehensively documented. Additionally, we have included more representative examples in **new Figure 10** of the revised manuscript.
>
> ---
>
> ### 5. Effect of Out-of-Distribution Images
>
> > **Question:** *"Does using the database as an anchor point affect the results for images that fall outside the database distribution?"*
>
> **Response:**
> No, it does not negatively affect the results. As detailed in our response to Section 2 (Minimax Target Selection) above, our method relies on repulsion, not retrieval.
>
> If an input image falls outside the database distribution (i.e., it is semantically distinct from the concepts in $\mathcal{E}$), it is mathematically **easier** to find a hard-negative prior with low cosine similarity. Therefore, "out-of-distribution" images actually provide a favorable condition for generating strong repulsive gradients, ensuring the effectiveness of the attack even for unseen semantic categories.
>
> ---
>
> ### 6. Clarification on Eq. (4) and Reconstruction Error
>
> > **Question:** *"Is the inclusion of min e in E intended to prevent the reconstructed error from deviating too far from the original images?"*
>
> **Response:**
> No. We wish to clarify the roles of these two equations to avoid any ambiguity.
>
> The term the reviewer referred to is part of the Target Selection objective in Equation (4). Its sole purpose is to act as a semantic selector: it calculates which concept in our database is most "opposite" to the input image to guide the attack. It is not responsible for visual similarity.
>
> The mechanism that actually "prevents the reconstructed error from deviating too far" is the perturbation budget constraint in Equation (5). This equation strictly limits the magnitude of the added noise (denoted as $\delta_k$) to be smaller than a threshold ($\epsilon$).
>
> Therefore, the logic is distinct: Equation (4) maximizes semantic disruption (making the model confused), while Equation (5) independently guarantees visual quality (keeping the image looking normal).
>
> ---
>
> ### 7. Elaboration on $N_{\text{max}}$ Selection
>
> > **Question:** *"Could the authors elaborate further on the selection of N_max and its effect on different classes?"*
>
> **Response:**
> Please refer to our detailed response in **Part 2 (Hyperparameter Choice and Trade-offs)** above. There, we explicitly analyze how increasing $N_{\text{max}}$ to 64 shifts the attack focus to fine-grained details, maximizing Tract/Block protection while sacrificing Metro-level performance—a trade-off aligned with our anti-doxing objective.
>
> ---
>
> We hope these responses fully address your concerns and clarify the methodological design of our work.

---

> > ### Comment · Reviewer_6ECh · 2025-11-20
> > **Response to Author's Rebuttal**
> >
> > I appreciate the authors’ detailed response. All of my concerns have been addressed, so I’ve made the following changes:
> >
> > * Presentation score: 2 $\rightarrow$ 3
> > * Rating: 4 $\rightarrow$ 6
> > * Confidence: 3 $\rightarrow$ 4
> >
> > That said, I still recommend adding the exact "concept" shown in Figure 3 directly to the paper. Since most readers will not see the rebuttal, the authors may not have another opportunity to clarify this point. From my perspective, including this concept is essential for understanding the rest of the paper, as it plays a central role in the overall methodology.
> >
> > Additionally, I have a few suggestions:
> >
> > * Please include the reasoning for choosing $N_{max}$, as explained in the third rebuttal response. This will help readers better understand how the hyperparameters are selected.
> > * From the authors’ explanation, metro or region seems less critical to the results. In the future, it may be worth exploring a more fine-grained approach that directly targets block or tract levels, since those details already subsume region and metro information.
> >
> > I hope these suggestions help the authors further strengthen the paper.

---

> > > ### Author Response · Authors · 2025-11-20
> > > **Many Thanks for the Swift Response and Score Increase**
> > >
> > > We are thrilled by your swift feedback and the score raise!
> > >
> > > We fully agree with your suggestions. Since we still have about two weeks in the discussion period, we will take our time to carefully figure out the best way to integrate these updates and ensure the paper is as clear as possible for future readers.
> > >
> > > Thanks again for your support.

---

### Official Review · Reviewer_MzBe · 2025-10-28

**Soundness:** 3
**Presentation:** 3
**Contribution:** 3
**Rating:** 6
**Confidence:** 3

**Summary:**

This paper presents ReasonBreak, an innovative adversarial framework that effectively counters geographic privacy threats posed by Multi-modal Large Reasoning Models (MLRMs) by disrupting their hierarchical reasoning processes through concept-aware perturbations. Unlike conventional methods that apply uniform noise, our approach strategically targets critical conceptual dependencies within reasoning chains, causing cascading failures in location inference. The framework is enabled by GeoPrivacy-6K, a comprehensive dataset of 6,341 ultra-high-resolution images with hierarchical annotations. Extensive evaluation across seven state-of-the-art MLRMs demonstrates remarkable effectiveness, achieving 33.8% tract-level and 33.5% block-level protection rates - nearly doubling baseline performance and establishing a new paradigm for reasoning-aware privacy defense against sophisticated AI threats.

**Strengths:**

1.The paper introduces the first concept-aware adversarial framework, ReasonBreak, specifically designed to target the hierarchical reasoning chains of multimodal large models. It achieves a paradigm shift in defense from the perception layer to the reasoning layer through a minimax target selection strategy.
2.The black-box transfer capability of this framework has been successfully validated on commercial APIs, including state-of-the-art models such as GPT-5 and Gemini 2.5 Pro, demonstrating its practical effectiveness in real-world scenarios.

**Weaknesses:**

1.The scenario of the paper is very clear. However, it does not clearly point out the technical difficulties of the problems in this scenario. It is hoped that the problems under this scenario will be analyzed.
2.It is recommended to additionally supplement a module diagram in the framework overview of Section 4.3 on page 5 to visually display the ReasonBreak framework.
3.The author only conducted training on the datasets they proposed and performed black-box testing on open-source datasets. It is recommended that the author provide white-box experimental results.
4.The author states that the training set used has a high resolution. What would be the impact if a low-resolution dataset were used?

**Questions:**

1.The scenario of the paper is very clear. However, it does not clearly point out the technical difficulties of the problems in this scenario. It is hoped that the problems under this scenario will be analyzed.
2.It is recommended to additionally supplement a module diagram in the framework overview of Section 4.3 on page 5 to visually display the ReasonBreak framework.
3.The author only conducted training on the datasets they proposed and performed black-box testing on open-source datasets. It is recommended that the author provide white-box experimental results.
4.The author states that the training set used has a high resolution. What would be the impact if a low-resolution dataset were used?

---

> ### Author Response · Authors · 2025-11-20
>
> **Part 1**
>
> We sincerely thank for your positive assessment and recognizing ReasonBreak as an "innovative adversarial framework" that achieves a "paradigm shift" in privacy defense. We appreciate the constructive feedback regarding the technical challenges analysis and experimental settings. Revisions in the PDF are highlighted in blue for clarity. Below, we address each comment in detail.
>
> ---
>
> ### Q1: Analysis of technical difficulties in the scenario
>
> > **Comment:** *"The scenario of the paper is very clear. However, it does not clearly point out the technical difficulties... It is hoped that the problems under this scenario will be analyzed."*
>
> **Response:**
> We appreciate this suggestion. The core technical challenge stems from the fundamental mismatch between the targets of traditional adversarial attacks and the evidence required for MLRM reasoning.
>
> In standard scenarios, such as an image of a dog on a lawn, conventional attacks typically maximize feature distortion on the most salient foreground object (the dog) to mislead classification. However, MLRMs function like visual detectives: they often bypass these dominant foreground features to proactively seek out fine-grained, non-salient background details—such as a specific tree species or a small storefront sign shown in Figure 1—to deduce location. Standard global perturbations tend to overlook these subtle "low-attention" regions.
>
> Consequently, the primary difficulty lies in redirecting the adversarial budget away from visually prominent features to targeted, often minute, semantic concepts that serve as the critical evidence nodes in the model's hierarchical reasoning chain.
>
> ### Q2: Request for a framework diagram
>
> > **Comment:** *"It is recommended to additionally supplement a module diagram in the framework overview of Section 4.3 on page 5 to visually display the ReasonBreak framework."*
>
> **Response:**
> We thank you for this helpful suggestion, as a visual representation significantly improves the clarity of our method. In the revised manuscript, we have added the requested diagram as **Figure 3** in Section 4.3.
>
> This figure provides an end-to-end illustration of the ReasonBreak pipeline, starting from the Adaptive Decomposition where the image is partitioned into blocks, followed by the Concept Assignment step which links spatial regions to specific geographic concepts. It further visually demonstrates our Minimax Target Selection strategy, showing how we identify a hard-negative prior from the embedding bank to guide the adversarial direction, and finally depicts the Perturbation Synthesis via the learnable decoder to reconstruct the privacy-protected image. We hope this new visualization makes the overall workflow more intuitive.

---

> ### Author Response · Authors · 2025-11-20
>
> **Part 2**
>
> ### Q3: Request for white-box experimental results
>
> > **Comment:** *"The author only conducted training on the datasets they proposed and performed black-box testing on open-source datasets. It is recommended that the author provide white-box experimental results."*
>
> **Response:**
> We thank you for this recommendation and appreciate the opportunity to clarify our experimental design. First, we would like to explicitly distinguish the roles and characteristics of the two datasets mentioned:
>
> * **DoxBench (Testing Set):** This is the "open-source dataset" specifically curated for evaluation. Crucially, DoxBench provides precise GPS coordinates (street-level) as ground truth, enabling the calculation of geolocation accuracy and privacy protection rates.
> * **GeoPrivacy-6K (Training Set):** This is the "dataset we proposed," which serves exclusively as a training resource. As illustrated in Figure 1 and further visualized in Appendix I, GeoPrivacy-6K is constructed with dense, hierarchical visual concept annotations to supervise the learning of concept disruption, but it **lacks the specific geographic coordinates** found in DoxBench.
>
> Without these ground-truth coordinates, it is technically infeasible to calculate the requested performance metrics (like distance error) on GeoPrivacy-6K. Consequently, our experiments are designed to train on the concept-rich GeoPrivacy-6K and evaluate the transferability of the attack on the coordinate-rich DoxBench.
>
> ### Q4: Impact of low-resolution datasets
>
> > **Comment:** *"The author states that the training set used has a high resolution. What would be the impact if a low-resolution dataset were used?"*
>
> **Response:**
> We appreciate you raising this point regarding the dataset characteristics. We respectfully clarify that the selection of high-resolution images for training is not a subjective preference, but an objective adaptation to the current real-world situation. Current mainstream social media platforms typically support kilopixel-level resolutions even after scaling (e.g., Instagram scales to 1080px width, while Facebook/Meta supports up to 2048px).
>
> The specific impact of using a low-resolution dataset for training would be a fundamental misalignment between the adversarial learning process and the reasoning capabilities of MLRMs. As identified in the DoxBench study, these models rely heavily on high-frequency visual cues—such as text on distant street signs or specific architectural textures—to perform precise localization. If we were to train our generator on low-resolution data, these critical fine-grained details would be effectively obliterated during downsampling. Consequently, the generator would be limited to learning perturbations on coarse, low-frequency features (e.g., global shapes or color histograms). When such a model is deployed against high-resolution targets, it would fail to mask the minute but decisive evidence that MLRMs actually exploit, rendering the defense ineffective against the state-of-the-art threats we aim to counter.
>
>
> ---
>
> We hope these clarifications regarding the technical challenges, framework visualization, and experimental design fully address your concerns.

---

### Official Review · Reviewer_XjM5 · 2025-10-30

**Soundness:** 2
**Presentation:** 3
**Contribution:** 3
**Rating:** 6
**Confidence:** 2

**Summary:**

This paper studies the problem of protecting the geographic privacy of images due to the ability of multimodal large reasoning models (MLRMs) to perform image geolocation. The authors first create a new GeoPrivacy-6K dataset consisting of high-resolution images of locations and bounding-box-based concept or feature annotations that are used during the defense strategy. The proposed defense ReasonBreak is motivated by the supposed step-by-step geolocation reasoning of reasoning models in inferring geolocation. If perturbations can affect any step in this reasoning process, the disruption can cascade to cause incorrect geolocation outputs from MLLMs. ReasonBreak works by identifying relevant concepts in each block of the image, and then finding a suitable perturbation that pushes the embedding of the perturbed block far from the original concepts it contained. The authors conduct an analysis of the effectiveness of their method across numerous open and closed source models and note its defensive effectiveness against baselines. Finally, the authors also perform ablations on the perturbation budget, block size, and the use of their target selection algorithm.

**Strengths:**

1. The paper introduces a novel method to disrupt the geolocation reasoning of MLRMs. To my knowledge, this is the first paper that explicitly targets the reasoning ability of MLRMs, by focusing on especially important concepts used in the reasoning process. It seems like this general paradigm would be extended outside of the domain of image geolocation to other intensive MLLM reasoning tasks.

2. ReasonBreak seems to significantly outperform the more general-purpose attack baselines, while remaining imperceptible, which is important for image geolocation, especially, i.e., many real-world attack use-cases may use social media images.

3. ReasonBreak does not require model weights to create defenses. This is a significant benefit, given that the most powerful and accessible MLRMs are closed source.

**Weaknesses:**

1. The theoretical motivation is a bit confusing. In lines 208-209, you mention that the model "progressively refines its location," and in lines 235 - 236, you mention that "an error introduced at an early state ... does not remain localized". While there have been methods that induce this sort of least to most reasoning for image geolocation through prompting [1], generally, R1-style reasoning models can backtrack and revisit conclusions made earlier in their reasoning trace.

2. The dataset construction, especially the scene annotation phase, is missing quite a few details. Did a model perform the annotation? If so, what model and what was the prompt for the model? How are bounding boxes identified by a model if it is used? It seems like the method requires similar concept annotations on any inference image, is this true? If so, how expensive are these image annotations for each image? Do you have an ablation on the model used for concept annotation with overall performance i.e., it would be great if this works well even with an open source VLM annotator?

3. Some parts of the method are unclear / could use an improvement in their description. For instance, the definition of N_max has to be inferred from the partitioning in equation 1. It would be good to define this clearly since there is a later ablation on this variable. The concept assignment section is also a bit confusing. What is the default concept? Does it have a specifically chosen embedding? How much overlap does there need to be between a concept and a block for the concept to be assigned?

4. The training set images are high resolution. Why was the choice made? It seems many social media images may be highly compressed, and these are often the images that adversaries may want to geolocate for malicious purposes. It would be necessary to verify that the method also works on images that are not high resolution in other geolocation datasets like IM2GPS.

5. JPEG compression is a known workaround for adversarial attacks on vision models. Does your defense hold up to JPEG compression?

6. While the current approach works well for MLRMs, it would be interesting to see if it can also work for specifically designed image geolocation models like [2].

[1] Mendes, Ethan, Yang Chen, James Hays, Sauvik Das, Wei Xu, and Alan Ritter. "Granular privacy control for geolocation with vision language models." arXiv preprint arXiv:2407.04952 (2024).

[2] Haas, Lukas, Michal Skreta, Silas Alberti, and Chelsea Finn. "Pigeon: Predicting image geolocations." In Proceedings of the IEEE/CVF Conference on Computer Vision and Pattern Recognition, pp. 12893-12902. 2024.

**Questions:**

1. If you do not use the ensemble training, how diminished are the results on various MLRMs?

2. Is there a correlation between protection with ReasonBreak and how famous or well-known an input image is? For instance, can you successfully protect the location of a famous landmark like the Eiffel Tower?

3. Prior work found that universally transferable image jailbreaks are hard to find [1]. How do you think your work fits in with these claims?


[1] Schaeffer, Rylan, Dan Valentine, Luke Bailey, James Chua, Cristobal Eyzaguirre, Zane Durante, Joe Benton et al. "Failures to find transferable image jailbreaks between vision-language models." arXiv preprint arXiv:2407.15211 (2024).

---

> ### Author Response · Authors · 2025-11-20
>
> **Part 1**
>
> We thank the reviewer for their constructive feedback and for identifying ReasonBreak as the "first paper that explicitly targets the reasoning ability of MLRMs." We are encouraged that this perspective resonates strongly across the review board. We appreciate that you also recognized our method's significant performance advantage over baselines.
>
> In response to your specific comments on practicality, we have expanded our evaluation with aggressive compression stress tests and efficiency benchmarks. These new results further confirm the robustness and deployment readiness of ReasonBreak. Revisions in the PDF are highlighted in blue for clarity.
>
> ---
>
> ### 1. Theoretical Motivation: Robustness to Backtracking
>
> > **Comment:** *"Reasoning models can backtrack and revisit conclusions... theoretical motivation is a bit confusing."*
>
> **Response:**
> This is an insightful observation regarding the non-linear nature of modern reasoning models. We agree that CoT allows for backtracking; however, our defense remains effective because it targets the visual axioms (primary evidence) rather than the reasoning path itself. ReasonBreak operates by corrupting the fundamental visual representation of geographic concepts, effectively establishing a "defense in depth" across the concept hierarchy.
>
> Consider a specific example: if a reasoning chain relies on a "Palm Tree" to infer a tropical region, ReasonBreak perturbs these specific visual features. Even if the model backtracks to verify the premise (*"Is this really a tropical setting?"*), it re-examines the image only to find the same corrupted evidence. Since our method simultaneously targets concepts across multiple scales—from local "street signs" to regional "vegetation"—backtracking essentially leads the model from one corrupted node to another. This creates a systemic collapse where self-correction mechanisms lack valid anchor points to recover the correct location.
>
> ---
>
> ### 2. Clarification on Annotation Details and Cost
>
> > **Comment:** *"The dataset construction... is missing quite a few details... Did a model perform the annotation? ... it would be great if this works well even with an open source VLM annotator?"*
>
> **Response:**
> We appreciate the reviewer’s detailed inquiries regarding the dataset construction and annotation pipeline. We wish to highlight that the specific details regarding the model architecture, prompt design, and coordinate specifications were comprehensively documented in Appendix B of the original submission due to space constraints in the main text. However, we are happy to elaborate here to fully resolve your concerns.
>
> The reviewer perceptively noted that *"it would be great if this works well even with an open source VLM annotator"*—we are pleased to confirm that our framework is indeed built upon exactly such a foundation. The entire annotation process is fully automated by **QwenVL 2.5 72B**, a state-of-the-art open-source model selected specifically for its superior spatial reasoning capabilities. This choice ensures the method is highly practical. Furthermore, we commit to releasing all these generated hierarchical annotations along with the entire GeoPrivacy-6K dataset.
>
> Regarding the specific mechanics, we designed a rigorous three-stage prompting pipeline—encompassing geographic filtering, hierarchical categorization, and reasoning chain extraction—where the exact prompt templates for each stage are explicitly provided in Appendix B.1. In the final stage, the model is instructed to output normalized square bounding boxes (`[center_x, center_y, size]`) alongside each identified concept. To further enhance clarity and provide intuitive verification of this process, beyond the textual descriptions originally provided, we have added representative visual examples of these automated annotations in the revised **Appendix Qualitative Examples of GeoPrivacy-6K (Figure 10)**. These visualizations demonstrate the precision of the open-source VLM in capturing fine-grained geographic cues without human intervention.

---

> ### Author Response · Authors · 2025-11-20
>
> **Part 2**
> ### 3. Methodological Clarifications ($N_{\text{max}}$ and Concept Assignment)
>
> > **Comment:** *"Some parts of the method are unclear... definition of N_max... concept assignment... What is the default concept?"*
>
> **Response:**
> We appreciate the reviewer’s scrutiny regarding the method definitions. We acknowledge that the original textual description was not sufficiently precise. To address this, we have **rewritten Section 4.3** to formalize these mechanisms and added a new **Figure 3** to visually illustrate the entire pipeline.
>
> * **$N_{\text{max}}$ (Adaptive Decomposition):** As clarified in the revised text, $N_{\text{max}}$ is a hyperparameter that strictly bounds the grid granularity. We introduced a formal optimization objective (Eq. 3) which dynamically finds the optimal grid dimensions $(m, n)$ to match the original image's aspect ratio while satisfying $mn \leq N_{\text{max}}$. This ensures fine-grained coverage while minimizing geometric distortion.
>
> * **Concept Assignment Protocol:** We have explicitly defined the assignment logic based on spatial intersection. As visualized in the new Figure 3, there are three possible outcomes of the concept assignment logic: a block may be assigned a single concept, multiple concepts, or the default set of all image concepts if it has no spatial overlap.
>
> * **Default Concept Strategy:** Regarding your specific question on "default concepts" for blocks without annotations (e.g., sky or road): we now clarify that these blocks receive the complete set of image concepts. This "conservative assignment" strategy is a deliberate design choice to ensure that even background regions are adversarially perturbed, preventing the model from exploiting global context for inference.
>
> ---
>
> ### 4. Impact of Ensemble Training
>
> > **Comment :** *"If you do not use the ensemble training, how diminished are the results on various MLRMs?"*
>
> **Response:**
> We thank the reviewer for this insightful question regarding the ablation of our training strategy.
>
> First, we note that ensemble training is the established standard for transfer-based adversarial attacks. All state-of-the-art baselines compared in our paper (e.g., M-Attack) similarly employ ensemble surrogates as their default configuration to ensure robust black-box transferability.
>
> To quantify the specific gain in our framework, we conducted an ablation study on InternVL 3.0 72B, varying the ensemble size from 1 to 4 (default).
>
> | Settings | Region | Metro. | Tract | Block |
> | :--- | :---: | :---: | :---: | :---: |
> | Ensemble (4 models) [Default] | 10.8 | 0.0 | 33.3 | 58.3 |
> | Ensemble (3 models) | 13.5 | 0.0 | 36.7 | 50.0 |
> | Ensemble (2 models) | 10.1 | 0.0 | 30.0 | 50.0 |
> | Single Model (1 model) | 4.2 | 0.0 | 30.0 | 41.7 |
>
> **Analysis:** As expected, the ensemble strategy is effective, boosting the critical Block-level protection from 41.7% to 58.3% by preventing overfitting to a single surrogate's feature space. However, it is worth highlighting that even with a single surrogate, ReasonBreak maintains a remarkable 41.7% protection rate at the Block level—still significantly outperforming the strongest baseline which utilized its own ensemble. This confirms that the efficacy of our method stems fundamentally from the concept-aware targeting mechanism, with ensemble training serving as a powerful amplifier rather than a prerequisite.

---

> ### Author Response · Authors · 2025-11-20
>
> **Part 3**
>
> ### 5. Justification for High-Resolution Images vs. IM2GPS
>
> > **Comment:** *"The training set images are high resolution. Why was the choice made? ... verify that the method also works on... IM2GPS."*
>
> **Response:**
> We appreciate the reviewer raising this point regarding the dataset characteristics. We respectfully clarify that the selection of high-resolution images is not a subjective preference, but an objective adaptation to the current real-world situation. Current mainstream social media platforms typically utilize kilopixel-level resolutions even after scaling (e.g., Instagram scales to 1080px width, while Facebook/Meta uses 2048px). The DoxBench study specifically identified that these high-resolution images (compared to the traditional $224 \times 224$ resolution in computer vision) contain significantly richer visual cues that facilitate precise localization by MLRMs. Our work follows this finding to address this specific privacy leakage issue; therefore, we collected a training dataset completely non-overlapping with DoxBench to verify our method's effectiveness.
>
> Regarding the lower-resolution images mentioned by the reviewer (e.g., those in IM2GPS), except for some famous landmarks, it is difficult for MLRMs to pinpoint them to street-level precision like the images in DoxBench. Consequently, the immediate privacy threat is significantly lower in those scenarios compared to the high-resolution social media context we target.
>
> Finally, concerning the image compression technologies used by social media, we emphasize that we have actually used standard JPEG compression by default in our main text experiments to simulate the real world. We will discuss this further in our subsequent responses below (including results on specialized geographic reasoning models) to help deeply understand the unique challenges brought by multi-modal reasoning models.
>
> ---
>
> ### 6. Robustness to JPEG Compression
>
> > **Comment:** *"JPEG compression is a known workaround for adversarial attacks on vision models. Does your defense hold up to JPEG compression?"*
>
> **Response:**
> We thank the reviewer for raising this critical point regarding defense robustness.
>
> First, we wish to clarify that all experimental results reported in the main text were already conducted using a standard JPEG quality factor ($Q$) of 95 to simulate a realistic baseline, rather than using uncompressed formats.
>
> To further address your concern about more aggressive compression, we performed a stress test (added in Appendix D) by reducing the quality factor to $Q=75$ and $Q=50$ using InternVL 3.0 72B.
>
> | Quality Factor ($Q$) | Region | Metro. | Tract | Block |
> | :--- | :---: | :---: | :---: | :---: |
> | $Q=95$ (Default) | 10.8 | 0.0 | 33.3 | 58.3 |
> | $Q=75$ | 11.1 | 0.0 | 33.3 | 58.3 |
> | $Q=50$ | 9.3 | 0.0 | 30.0 | 58.3 |
>
> **Analysis:** As shown in the table, ReasonBreak exhibits remarkable stability. Reducing the quality factor results in virtually no degradation in the critical Block-level protection (maintaining 58.3% even at $Q=50$). This resilience suggests that our concept-aware perturbations are structurally robust and effective against standard social media image processing pipelines.

---

> ### Author Response · Authors · 2025-11-20
>
> **Part 4**
>
> ### 7. Comparison with Specialized Geolocation Models
>
> > **Comment:** *"While the current approach works well for MLRMs, it would be interesting to see if it can also work for specifically designed image geolocation models like [PIGEON] [2]."*
>
> **Response:**
> We thank the reviewer for this valuable suggestion. We attempted to evaluate PIGEON [2]; however, we were unable to proceed as the official repository explicitly states: *"This repository is purely meant for the academic validation of the paper's code... model weights are not provided as part of this release."* Given this constraint, we instead evaluated **GeoCLIP** (Vivanco et al.), a comparable state-of-the-art specialized geolocation model, on the DoxBench dataset.
>
> The results empirically validate the core premise of our work. GeoCLIP achieved a high accuracy of 94.8% at the coarse Region level and 62.0% at the Metropolitan level, demonstrating its proficiency in global retrieval. However, its performance collapsed to 0.0% at the fine-grained Tract and Block levels.
>
> | Metric | Region | Metro. | Tract | Block |
> | :--- | :---: | :---: | :---: | :---: |
> | **GeoCLIP Accuracy** | 94.8% | 62.0% | 0.0% | 0.0% |
>
> This sharp contrast highlights a fundamental distinction in the threat landscape: while specialized models excel at recognizing general geographic regions via global visual alignment, they lack the hierarchical reasoning capabilities required to pinpoint street-level locations from fine-grained details. Consequently, for the critical privacy metrics of Tract and Block levels, traditional specialized models pose little threat as they naturally fail to infer such precision. This finding reinforces the conclusion of DoxBench: MLRMs represent a unique privacy vulnerability precisely because they possess the reasoning capacity to bridge the gap from a general region to a specific street block. ReasonBreak is thus specifically engineered to counter this high-stakes reasoning capability.
>
> *Vivanco Cepeda et al., Geoclip: Clip-inspired alignment between locations and images for effective worldwide geo-localization, NeurIPS 2023*
>
> ---
>
> ### 8. Correlation with "Famous" Images
>
> > **Comment:** *"Is there a correlation between protection with ReasonBreak and how famous or well-known an input image is? e.g., can you successfully protect the location of a famous landmark like the Eiffel Tower?"*
>
> **Response:**
> We thank the reviewer for this interesting question. It inspired us to conduct a qualitative failure analysis (Section 5.5), and the results align with your intuition to some extent, though with a specific nuance.
>
> We found that highly distinct identifiers do challenge our defense, but the primary failure mode we observed involves **explicit, high-saliency text** rather than visual fame alone. For instance, as shown in the new **Figure 6**, our method struggled against a sign explicitly reading "1565, B46, Google". In such cases, it appears the model simply "reads" the answer via OCR rather than reasoning about the scene.
>
> This suggests an interesting distinction: while we cannot verify this on the Eiffel Tower specifically (as it is not in DoxBench), our results imply that the boundary seems to lie less in the "fame" of the location, and more in whether the recognition relies on visual reasoning or direct textual reading.
>
> ---
>
> ### 9. Relation to "Failures to Find Transferable Image Jailbreaks" [1]
>
> > **Comment:** *Prior work found that universally transferable image jailbreaks are hard to find [1]. How do you think your work fits in with these claims?*
>
> **Response:**
> We appreciate the reviewer connecting our work to this important recent study (Schaeffer et al., 2024). We believe there is no contradiction, as the distinction lies in the fundamental objective of the attack: **Jailbreaking vs. Visual Disruption**.
>
> The study [1] focuses on "jailbreaks"—adversarial inputs designed to bypass safety refusals and force models to generate specific prohibited content. This is inherently difficult to transfer because it requires the perturbation to precisely steer the model's latent representation into a highly specific, often "guarded" semantic region (the safety guardrails).
>
> In contrast, ReasonBreak targets **privacy protection**, which relies on disrupting visual perception. Our goal is not to force a specific malicious output, but merely to invalidate the visual evidence required for correct recognition (e.g., making a "palm tree" unrecognizable). As established in the broader adversarial learning literature, disrupting perceptual features is a significantly more transferable task than inducing specific behavioral control. To clarify this landscape for readers, we have explicitly incorporated a discussion of [1] into the opening of our **Related Work** section in the revised paper, distinguishing the challenges of directed jailbreaks from the transferability of perceptual attacks.
>
> ---
>
> We hope these responses fully address your concerns.

---

> > ### Comment · Reviewer_XjM5 · 2025-11-21
> >
> > Thank you for your response and for updating the paper. I find that the response, updated discussion, and new results have improved the paper. I think this paper is an important contribution to a new attack surface (VLM image geolocation), so I have updated my score to 8.
> >
> > However, while I understand your point about high-resolution images, I think the paper would benefit from evaluation on IM2GPS or some other common geolocation benchmark that has lower-resolution images.

---

### Meta-Review · Area_Chair_vWrF · 2026-01-09

**Summary:**

The paper introduces ReasonBreak, a novel concept-aware adversarial framework that disrupts the hierarchical geolocation reasoning of Multimodal Large Language Models (MLLMs) by strategically perturbing key visual concepts, achieving effective, imperceptible, and transferable black-box protection. Reviewers recognized the novel motivation, strong empirical results, and practical black-box efficacy. Key concerns include clarifying methodological details—such as the theoretical motivation for reasoning disruption, the dataset annotation process, and the definitions of core components (e.g., concept assignment)—and conducting supplemental experiments to test robustness against JPEG compression, lower-resolution images, and non-MLLM geolocation models, as well as providing a framework diagram and computational cost analysis.

**Reviewer Concerns:**

The reviewer concerns are well addressed by the author's rebuttal and the feedback from reviewers indicates positive opinions. Area Chair (AC) is in agreement with the positive assessment provided by the four reviewers.

**Reviewer Scores:**

All reviewers have reached a consensus on accepting this paper. Following a comprehensive evaluation of the reviewers' comments and the authors' rebuttal, the Area Chair concurs with this recommendation and endorses the paper for publication.

---

### Decision · Program_Chairs · 2026-01-26

Accept (Poster)